# Phasing analysis of lung cancer genomes using a long read sequencer

Yoshitaka Sakamoto[1,6], Shuhei Miyake[1,6], Miho Oka[1,2,6], Akinori Kanai[1], Yosuke Kawai [3], Satoi Nagasawa[1], Yuichi Shiraishi[4], Katsushi Tokunaga [3], Takashi Kohno [5], Masahide Seki [1], Yutaka Suzuki [1✉] & Ayako Suzuki [1✉]

Chromosomal backgrounds of cancerous mutations still remain elusive. Here, we conduct the phasing analysis of non-small cell lung cancer specimens of 20 Japanese patients. By the combinatory use of short and long read sequencing data, we obtain long phased blocks of 834 kb in N50 length with >99% concordance rate. By analyzing the obtained phasing information, we reveal that several cancer genomes harbor regions in which mutations are unevenly distributed to either of two haplotypes. Large-scale chromosomal rearrangement events, which resemble chromothripsis events but have smaller scales, occur on only one chromosome, and these events account for the observed biased distributions. Interestingly, the events are characteristic of *EGFR* mutation-positive lung adenocarcinomas. Further integration of long read epigenomic and transcriptomic data reveal that haploid chromosomes are not always at equivalent transcriptomic/epigenomic conditions. Distinct chromosomal backgrounds are responsible for later cancerous aberrations in a haplotype-specific manner.

[1] Department of Computational Biology and Medical Sciences, Graduate School of Frontier Sciences, The University of Tokyo, Chiba, Japan. [2] Ono Pharmaceutical Co., Ltd, Ibaraki, Japan. [3] Genome Medical Science Project (Toyama), National Center for Global Health and Medicine, Tokyo, Japan. [4] Division of Genome Analysis Platform Development, National Cancer Center Research Institute, Tokyo, Japan. [5] Division of Genome Biology, National Cancer Center Research Institute, Tokyo, Japan. [6] These authors contributed equally: Yoshitaka Sakamoto, Shuhei Miyake, Miho Oka. ✉email: ysuzuki@hgc.jp; asuzuki@edu.k.u-tokyo.ac.jp

Large-scale cancer genome studies have revealed numerous cancer-related mutations and identified key driver genes[1]. Several relevant drug targets and biomarkers have been identified, such as *EGFR* and *BRAF*[2–5]. So far, most studies have been conducted using short read sequencers. Therefore, our current knowledge has been limited mainly to mutations that occur in small-scale regions of genomes; the so-called single nucleotide variants (SNVs) and short insertions and deletions (indels).

Recently, larger genomic structural variants (SVs) have been identified in the genomes of various cancer types. These SVs are expected to have no less biological and clinical relevance. For example, both the chromosomal inversion and translocation generate oncogenic fusion genes, such as *BCR-ABL*[6], *EML4-ALK*[7], and *KIF5B-RET*[8]. In tumor-suppressor genes, such as *TP53*, *RB1*, and *PTEN*, large deletions frequently occur, thereby inactivating the expression and functions of these genes[9]. The Pan-Cancer Analysis of Whole Genomes Consortium has also focused on large-scale genomic aberrations in addition to SNVs. The consortium reported the SV signatures of 38 cancer subtypes[10]. Despite the potential relevance of SVs, conventional detection methods are based on short read sequencing data[11] and have limited validity toward the precise detection of SVs. In fact, the conventional analytical methodology may infer the presence of SVs but can only partially reveal their complete structures. To achieve a more direct and precise detection of SVs, long read sequencing should be employed for interrogating of various aspects of cancer genomes.

For this purpose, experimental and bioinformatics procedures for long read sequencing have recently recorded substantial progress. Although the fidelity of existing long read sequencing technologies remains ~90% for a single-pass read, several efforts have been collectively made to improve sequence accuracy[12]. For example, circular consensus sequencing has been developed as a means to construct more accurate sequences with 99% identity in the PacBio platform[13]. Recently, Oxford Nanopore Technologies (ONT) have announced the release of Q20 chemistry and base-calling system that enables single-pass sequencing with more than 99% accuracy. It is now realistic to use long read sequencers to systematically analyze a wider range of cancerous mutations, such as SNVs, relatively large-scale SVs and chromosomal-level rearrangements. In fact, several reports on the cancer genome long read analysis have recently revealed that, occasionally, newly discovered SVs demonstrate complex patterns of genomic aberrations[14–16].

Another unique advantage of employing long read sequencing for cancer genome analysis lies in its potential to reveal chromosomal contexts in which cancerous mutations are harbored[16]. Long read sequences should directly represent a mutual relationship between two mutations detected in the same read at a single-molecule level. This so-called "haplotype phasing analysis" would shed more light on a particular event occurring in a cancer type on either of the chromosomes of diploid genomes at a single molecule and haplotype resolution[17]. Each haplotype may reside in a distinct condition, which might be due to their differential DNA methylation or other epigenomic statuses possibly caused by the original lineage-specific regulations or other cancerous aberrant regulations at later steps[18]. Therefore, the consequentially occurring mutation patterns might serve as the footprints of the cancer genome evolution and could contain essential information for elucidating the causes and effects of mutations in the same cancer genomes. It is possible that a better understanding of such chromosomal contexts of cancerous mutations will shed new light on cancerous events for patient cases whose molecular etiology remains unknown from previous short read sequencing and provide a novel therapeutic insight.

In this study, we conduct a phasing analysis of cancer genomes combining short and long read sequencing technologies. We use whole-genome sequencing (WGS) data obtained from Japanese non-small cell lung cancer patients, where we identify a series of complex SVs[14]. We have further enriched sequencing depths for accurate phasing analysis and performed epigenome and transcriptome analyses. As such, we reveal the cancerous mutations from their chromosomal backgrounds' perspective. Here, we demonstrate that the obtained phasing results provide essential information for understanding the history of mutations and their possible causes.

## Results

**Phasing analysis of a lung cancer genome**. We performed our phasing analysis using the long and short read WGS data obtained from 20 non-small cell lung cancer specimens of Japanese patients. We constructed phased blocks using WhatsHap (version 1.0), which assumes the number of haplotypes as two. We used these data to elucidate the chromosomal backgrounds of the somatic SVs and SNVs at a single-haplotype level (Fig. 1a). In this section, we consider case S21 (having an L858R mutation in the *EGFR* as a driver mutation) as an example. For analyzing cancerous mutations, we first constructed the base phasing information for their matched-normal counterpart genome as a reference. We analyzed the whole-genome sequences of the PromethION data (57 Gb; 19×) of "case S21 normal (S21-N)." A bioinformatics scheme of the undertaken phasing analysis is provided in Supplementary Fig. S1. A total of 8816 phased blocks were generated with a median of 110-kb length; a length that is in agreement with previous reports[19,20]. We subsequently examined and found that the phased blocks harbored 1,880,725 heterozygous single nucleotide polymorphisms (SNPs) (Fig. 1b and Table 1). In total, 56% of the SNPs detected in this normal genome were assigned to the obtained phase blocks.

Then, we performed a similar phasing analysis of the tumor genome of "case S21 (S21-T)." The PromethION WGS data (105 Gb; 35×) were similarly phased using the germline SNPs detected in its normal genome. In total, 4300 phased blocks were successfully constructed. The median and maximum lengths were 268 kb and 8.5 Mb, respectively (Fig. 1c). An example of the performed tumor phasing analysis is presented in Fig. 1d. Afterward, somatic mutations were associated with the constructed haplotype blocks.

Briefly, the tumor blocks were somewhat longer in cancer genomes than in normal genomes. We considered that this difference might be because the sequencing depth was greater in the cancer genomes (35× and 19× for cancer and normal genomes, respectively). To examine this possibility, we examined the sequencing depth and phasing analysis results. Particularly, we examined the degree of the successful haplotype phasing at several sequencing depths (5×, 10×, 15×, 20×, 25×, 30×) and all reads by random sampling of the sequence reads (Fig. 1e). We found that the block length was unsaturated when using more than a 25× depth of the sequencing data. Moreover, the number of constructed blocks appeared to be saturated to around 5000 at a sequence depth of 20×–30×. Therefore, we considered that a depth of at least 20× for the sequencing data may not be perfect but should still be reasonable to start the tumor phasing analysis.

To investigate whether the obtained phasing results were sufficiently accurate, we evaluated the mutual relationship of the obtained haplotype blocks of the two given SNPs. We first compared the phased information on the tumor genome with that on the normal counterpart, which was separately constructed. We assumed that most germline SNPs should yield the

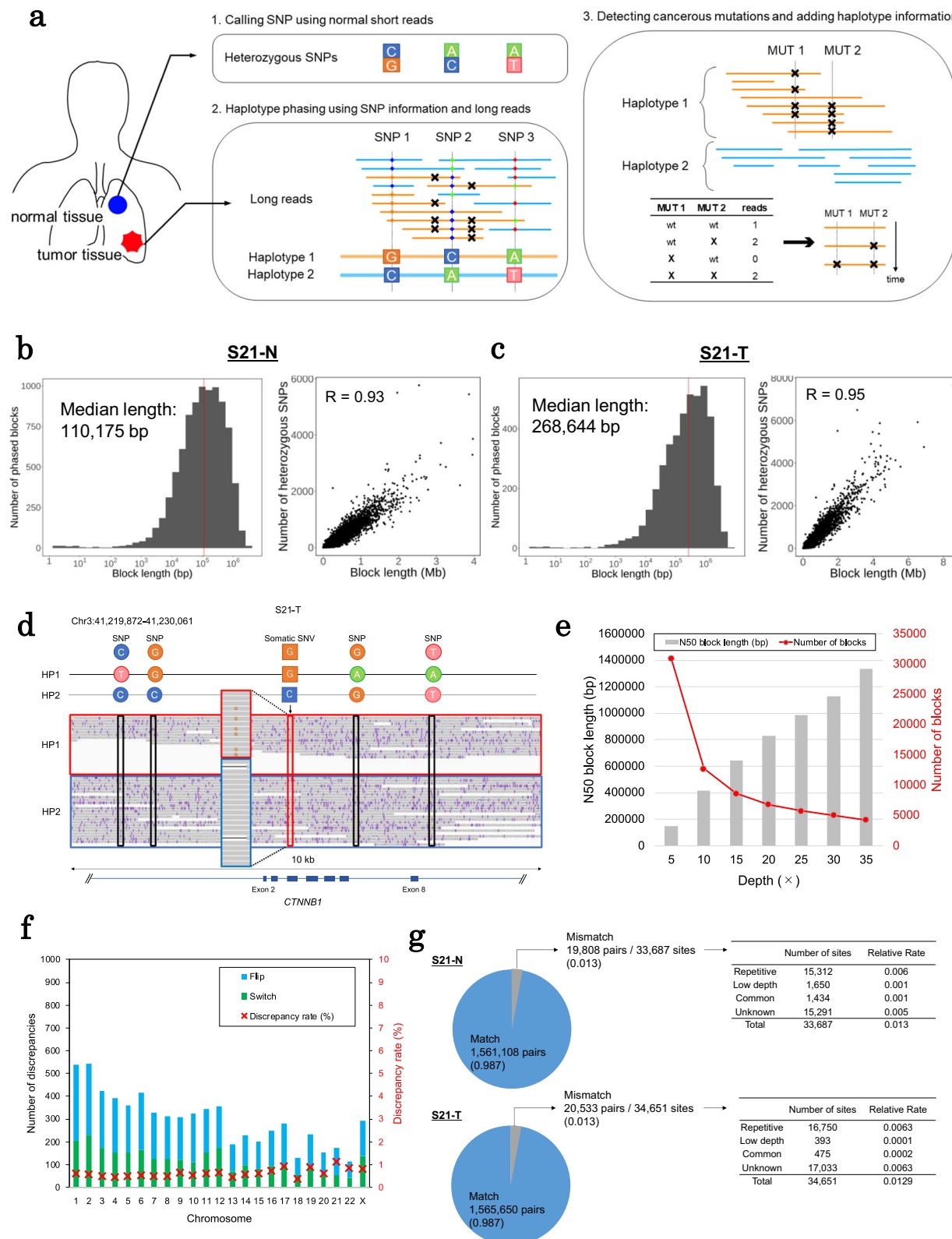

same association. The calculated discrepancy rates of the two SNPs between the tumor and normal genomes, including their switch and flip errors, appeared to be only 0.59%, which is a reasonable value considering a previous report[21] (Fig. 1f). We further compared our phasing results with those obtained from another study of a healthy Japanese cohort (Todai/Tokyo Health Control: THC). We considered haplotype information on 836

haplotypes obtained from 418 healthy individuals, as identified from the short read WGS data. The comparison showed that 98.7% of the SNP–SNP associations were consistent (Fig. 1g). These results indicate that the obtained phasing information from both tumor and normal genomes are reasonably precise and can be used as references for performing further analyses of the genomic mutations at the haplotype level.

**Fig. 1 Phasing results of the representative case, case S21. a** An overview of the haplotype phasing analysis is as follows. Step 1: The SNP information was obtained from the analysis using short read sequences of normal specimens. Step 2: This step was performed for normal and tumor specimens; long reads were mapped to the reference genome, and phasing analysis was subsequently performed using the SNP information obtained in Step 1. Step 3: For tumor specimens, cancer-specific mutations were detected and annotated with the haplotype information obtained in Step 2. The distributions of the number of phased blocks in each block length (left panel) and the association of the number of heterozygous SNPs included in the blocks and block lengths (right panel) of the normal (**b**) and tumor (**c**) genomes. **d** An example of tumor genome phasing in the *CTNNB1* gene of case S21 bearing a somatic SNV. HP1 and HP2 indicate haplotypes #1 and #2, respectively. **e** Association between the sequencing depths and the construction of phased blocks (N50 block length and the number of blocks). **f** The number of discrepancies and discrepancy rates of phase information between the normal and tumor genomes in case S21. The discrepancies include flip and switch errors. A flip error means that a SNP is assigned to the other haplotype. A switch error means a haplotype is switched to the other haplotype at a SNP in the middle of a haplotype block. **g** Comparison of phase information with THC (upper panel: S21 normal, lower panel: S21 tumor). Detailed descriptions of the mismatch sites are described in the provided tables.

## Table 1 A summary of the phasing analysis performed on case S21.

|  |  | Normal | Tumor |
|---|---|---|---|
| Sequencing (PromethION) | Total yields (Gb) | 57 | 105 |
|  | Depth (×) | 19 | 35 |
|  | N50 read length (bp) | 16,812 | 20,092 |
| Phasing | Number of blocks | 8816 | 4300 |
|  | N50 block length (bp) | 634,053 | 1,336,363 |
|  | Number of phased SNP | 1,880,725 | 1,882,776 |
|  | Coverage | 0.79 | 0.86 |

**Phasing analysis of 20 cancer genomes**. As shown for case S21, we similarly analyzed the normal and tumor genomes for 20 non-small cell lung cancers. An average of 9027 and 6536 phased blocks were generated in the normal and tumor genomes, respectively (Fig. 2a, b); the median N50 lengths of the phased blocks were 665,168 and 1,002,105 bp, containing 1,848,079 and 1,850,441 heterozygous SNPs on average, respectively (Supplementary Tables S1 and S2). Occasionally, short blocks, which were supposed to derive from the lower sequencing depth, the shorter read length, or both, were detected. To further inspect these features, we assessed the association between the obtained block lengths and sequencing depths or read lengths (Fig. 2c). The sequencing depth roughly showed a positive correlation with the phased block length (0.31 $R^2$ value from the linear regression), suggesting once again that sufficient sequencing depth is a crucial factor for obtaining long phased blocks. However, we also noticed that there were some cases for which the phased block lengths were shorter despite their superior sequencing depth (indicated by arrows in Fig. 2c). For example, the N50 read lengths of cases S11-T, S14-T, and S16-T were, respectively, 6865, 11,385, and 5957 bp, which are shorter than the average (16,556 bp) despite their sequencing depths being 39×, 38× and 34×, respectively, which are deeper than the average (32×) (Supplementary Tables S1 and S2). We further analyzed the association between the lengths of individual reads and those of the constructed phased blocks. A strong correlation was detected between them (Fig. 2d; 0.55 $R^2$ value from linear regression). We also confirmed that copy number (CN) aberrations, which were supposed to shorten blocks, did not affect block lengths (Supplementary Fig. S2). These results indicated that the length of individual reads should contribute more significantly to the resulting phasing block than the sequencing depth.

Similar to case S21, we evaluated the precise generation of the phased blocks for the other 19 cases. As shown in Supplementary Table S3, we found that the discrepancy rates among the normal and tumor genomes were very low (0.5–1.1%; 0.76% on average). Compared with the THC dataset, 98.7% of the SNP–SNP association was consistent in the 20 cases, on average (Supplementary Table S4). Notably, the discrepant cases should include the true haplotype diversities or the cancerous rearrangements of the SNPs; thus, the real false rates should be even lower. Further, we investigated whether the phased blocks of the tumor specimens could comprehensively cover the genomic regions (including the repetitive or "SNP-poor" regions). We found that an average of 78% of the genomic regions contributed to the phased blocks in all 20 cases. The remaining 22%, which could not be covered by the phased blocks (low coverage regions), were mostly from regions characterized by low heterozygous SNPs (Fig. 2e).

To further validate the obtained phased blocks, we employed the newly available PromethION Q20 platform, through which the long read sequencing is expected to achieve an accuracy of 99% for a single read. We first confirmed that the sequence fidelity reached 96% when using the Q20 platform via our reference template analysis, "Human DNA HG002" (Supplementary Fig. S3 and Supplementary Table S5). Having observed the initial successful sequencing analysis, we subsequently performed the WGS analysis using the Q20 platform for S20-T. We identified only 0.35–1.8% of inconsistent cases (Fig. 2f). We also found that, although the sequence fidelity improved significantly, this analysis had only a limited effect on the constructed phased block lengths. An example of the obtained phasing results comparing the current version and that of Q20 is presented in Fig. 2g. Considering these results, the generated phase information was sufficiently precise, although it started from sequencing data with ~90% sequence identity to the current PromethION platform.

**Characterization of cancerous mutations**. Before the biological interpretation of data, we further minutely inspected the identified cancerous mutations. In each specimen, mutations of various categories, such as SNVs and SVs, larger chromosomal rearrangements and CN aberrations, were identified. The constructed catalog of these mutations from each specimen is summarized in Supplementary Fig. S4. Overall, the results showed no obvious discrepancy with those identified from the short read analysis (see our previous study[14] for details).

Overall, we could find that 43% of the point mutations were mapped to either haplotype #1 (HP1) or haplotype #2 (HP2) (38% on average; pale blue circles; Fig. 3a, see also Supplementary Fig. S5). For four particular cases (S1, S15, S17, and S18), the haplotype assignment was successful for less than 15% of the mutations. These were also cases in which the average variant allele frequencies (VAFs) were generally lower, suggesting originally low cancer genome contents (Supplementary Fig. S5b). The mutations of low VAFs were occasionally not covered by sufficient read depth by the PromethION; thus, the precise separation of their haplotypes may have been inhibited. The tumor purity and heterogeneity of the tumor specimens

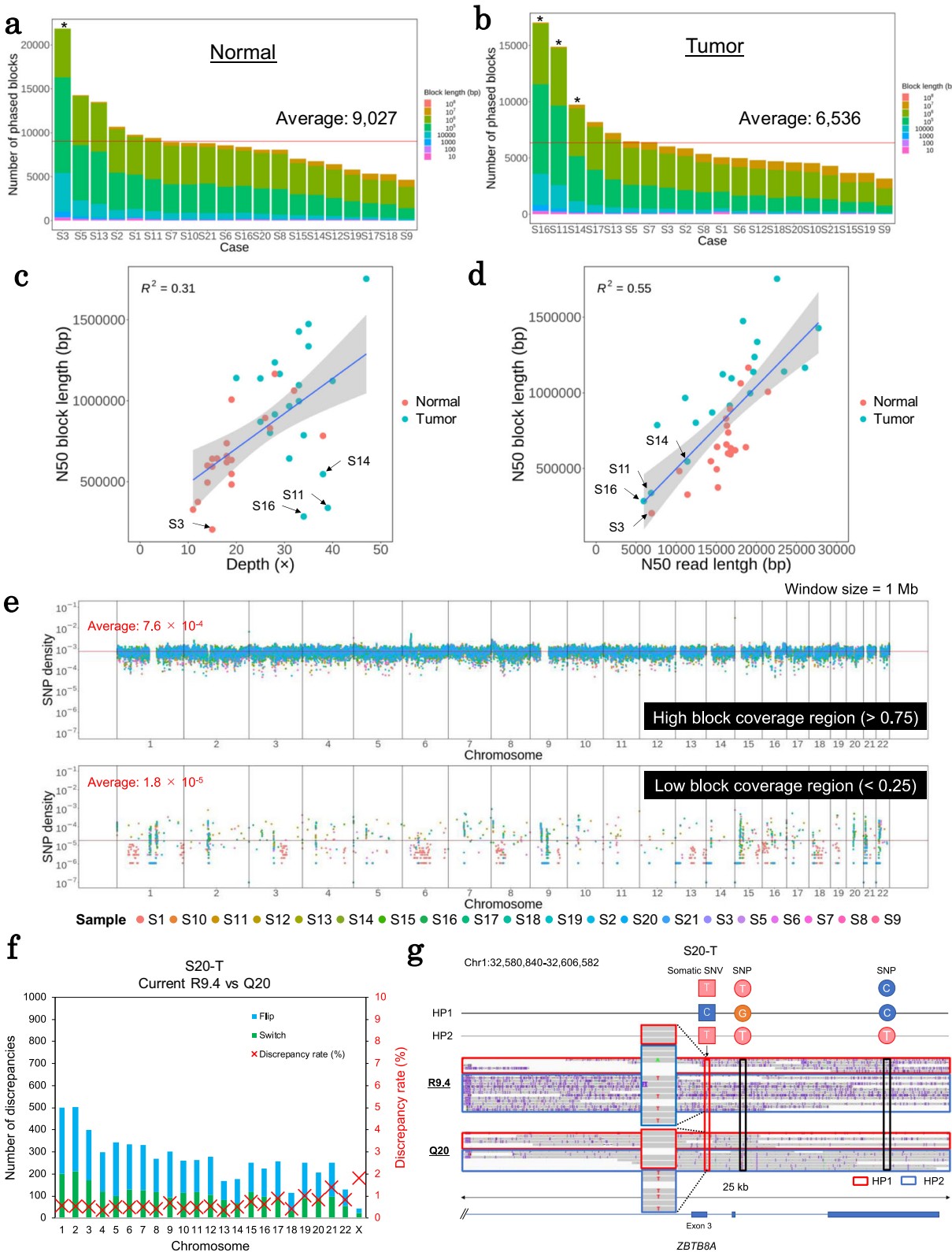

should also be considered crucial factors in ensuring sufficient sequencing depth; the latter is particularly crucial for assigning the cancerous mutations to the phased blocks. In fact, when we considered the mutations having VAFs of 0.2 or more, the total and average success rates of the assignment improved to 71% and 50%, respectively (dark blue circles; Fig. 3a).

We also validated the correct assignments of SVs. As a result, up to 70% of SVs, except for the inversion type, were successfully assigned to a single haplotype on phased blocks (Fig. 3b, also see Supplementary Figs. S4 and S5). As for the inversions, only up to 30% were phased. However, most unphased inversions (1669 of 2241) originated from case S16 (a large cell neuroendocrine carcinoma). The difficulty in phasing these inversions may lie in

**Fig. 2 Phasing results of the 20 cases.** The number of phased blocks in the normal (**a**) and tumor (**b**) genomes of the examined 20 cases. Red lines indicate the average numbers of phased blocks. Asterisks indicate the marked specimens in **c** and **d**. **c** Association between block lengths and depths. Cases S3 normal, S11 tumor, S14 tumor, and S16 tumor are identified far from the regression line. The shaded area indicates 0.95 confidence interval. **d** Association between the block length and read length. The shaded area indicates 0.95 confidence interval. **e** Genome-wide distribution of the SNP density with high (>0.75; upper panel) and low (<0.25; lower panel) coverage sites in a 1-Mb window in autosomal chromosomes. Red lines represent the averages of the SNP density. **f** Representation of the number of discrepancies and the discrepancy rates of phase information on case S20 tumor between the current R9.4 and Q20 PromethION platforms. **g** An example of phase information and an IGV image of the R9.4 and Q20 platforms of case S20, including a somatic mutation in the *ZBTB8A* gene. Q20 reads have fewer sequence errors than the R9.4 reads, and represent the SNPs and somatic SNVs precisely.

the block length's shortness of case S16 (Fig. 2b), which has probably harbored numerous inversion SVs.

**Biological and clinical inferences of the haplotype context of cancerous mutations.** After a series of technical validations, we attempted to elucidate the potential biological or clinical relevance of the detected cancerous mutations against their haplotype backgrounds. We just classified the somatic mutations into either of the two maternal/paternal haplotypes using haplotype-tagged reads. The relationship between somatic mutations with a distance longer than the reads were not analyzed (Supplementary Fig. S6). We first focused on mutations detected by the representative oncogenes and tumor-suppressor genes. Based on previous reports on non-small cell lung cancers, we selected eleven oncogenes, nine tumor-suppressor genes, and seven chromatin-remodeling and splicing-associated genes known to be frequently mutated in lung cancers[3]. The mutation patterns and their phased block information are illustrated in Fig. 3c.

In particular, the haplotype information about the mutations in the *EGFR* gene could be almost completely resolved (except for two indel cases) for all SNVs in all detected cases. For example, in case S21, two mutations were detected (L858R and R776S of *EGFR*), and both of them were mapped on the same haplotype in the same phased block (Fig. 3d). Precisely, 12 reads covered both mutations. All these reads represented either the wild-type or double mutants (except for two error reads; Supplementary Fig. S7a). Two possibilities were raised: either (i) these mutations have been acquired almost simultaneously or (ii) the double-mutant clones had somehow become more dominant than single mutants during cancer evolution. The *EGFR* L858R mutation is well-known as the most robust driver mutation, a reason it is well-known as a target for the EGFR tyrosine kinase inhibitor therapy. Moreover, there are a few reports describing the R776S mutation[22]. The R776S mutation may be the first to be introduced to the *EGFR* gene, and by providing a favorable backdrop or gaining some time, the L858R mutation was acquired as a result, leading to the strongest driver being those cells carrying both mutations. Future analyses of a larger number of cases could provide a relevant clue from the "predisposition" or "accessory" mutations' viewpoint, and might further elucidate the history of cancer genomes as to how the most robust cancer driver mutations may eventually prevail.

As an example of the phased SVs, case S14 harbored a large deletion in the *PTEN* gene. When one haplotype harboring an SV breakpoint in the *PTEN* gene (chr10:87,880,311) was designated as HP2, essentially almost no reads were mapped on HP1 in this same block (block#87809597). At the other end of the SV, the breakpoint (chr10:87,916,683) was connected to the HP1 of a different block (block#87913812) (Fig. 3e). We could detect three independent reads that spanned both ends of this SV, thereby connecting HP2 (block#87809597) to HP1 (block#87913812; see Supplementary Fig. S7b for more details). These results suggested that this gene region is a region of the loss of heterozygosity (LOH); one being the deletion between HP2 and HP1, and the

other being the deletion of a wider region. In a wider view of chromosome 10, the VAFs of the SNPs, which were heterozygous in its normal genome, were confirmed to retain almost 100% of the entire region (Supplementary Fig. S7c). These findings indicate that the LOH occurred in the region having the *PTEN* gene in the core, and the resultant mutant allele became dominant. Biologically, *PTEN* has a tumor-suppressor function, and its loss leads to the constant activation of the oncogenic signaling in the PI3K/AKT pathway. Interestingly, in case S14, another mutation, the *PIK3CA* E545K (which is a reported gain-of-function type mutation for the PI3K/AKT pathway), was detected (Supplementary Fig. S7d). Gene expression profiles, which characterized each case and were associated with the PI3K/AKT pathway, were confirmed by RNA-seq in case S14 for validation (Supplementary Fig. S8). The VAFs of this mutation were 18%, whereas the frequency of the *PTEN* loss was nearly 100%. Collectively, as for case S14, the genomic aberration of *PTEN* and the occurrence of LOH may have initially driven its carcinogenic process. During tumor progression, a yet additional *PIK3CA* mutation might have been acquired, thereby realizing the full activation of the PI3K/AKT pathway. The clonal structure of this case was also verified using PyClone-VI (Supplementary Fig. S9).

Moreover, for some cases, we could identify the occurrence order of multiple mutations occurring in different genes located in neighboring regions. In 20 cases, an average of 69 (0–649) mutation pairs could be ordered, for which their presence or absence was represented directly by individual long reads (Supplementary Fig. S9a). As exemplified in Fig. 3f, the sequence reads represented the presence or absence of the two mutations C > A in the *NKAIN4* 3'UTR (chr20:63,240,923, VAF: 0.42) and T > A in the *BIRC7* exon 6 (chr20:63,239,424, VAF: 0.25), in this order. We also validated the occurrence order of mutations by performing clonal structure inference using short read data and found that 44 mutation pairs were successfully assigned to different clones (Supplementary Table S6). Association between two mutations in the *PDGFD* intron of case S20 was shown as a successful example (Supplementary Fig. S9, single-cell DNA sequencing (scDNA-seq) was performed in two representative cases to further validate the inferred clonal structures; Supplementary Fig. S10 and Supplementary Table S7). However, a large fraction of the mutation pairs were still mapped to the same clones because the sequencing depths of the short read data were insufficient to resolve minor clones with low VAFs. Long read sequencing can more sensitively detect the occurrence of mutations adjacent to each other. By further extending the phased block length, we would eventually detect the order of cancerous mutations, although most of them are merely passenger mutations, and we would reconstruct the diverse evolutionary histories of individual cancers.

**Characterization of transcriptional aberrations at a haplotype level.** We next tried identifying the potential regulatory mutations, which, when present, should affect the aberrant

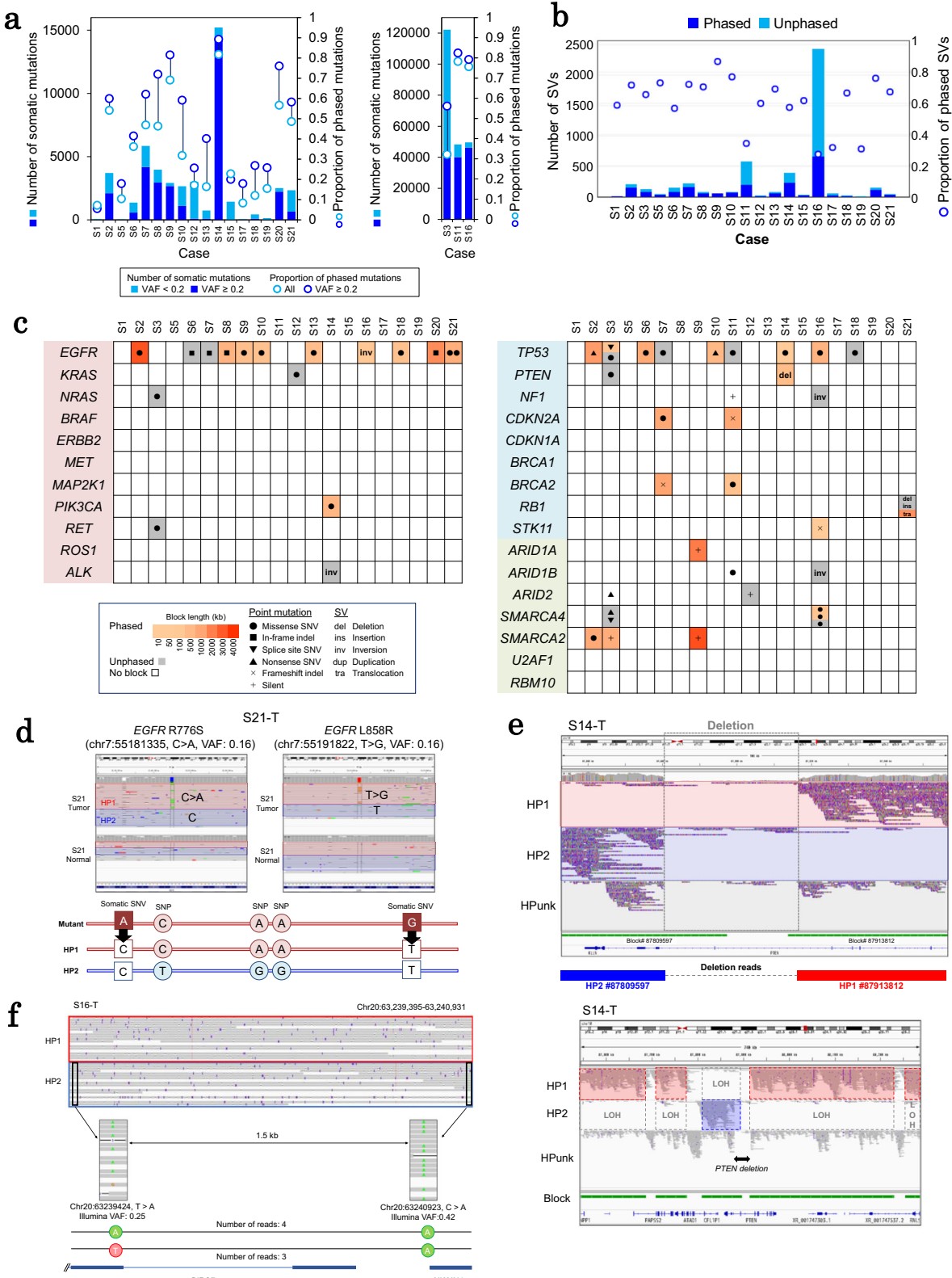

transcriptions in cancers[18]. At the genomic level, 33,820 regulatory mutation candidates were located in either the promoter or enhancer regions according to previous studies[23], and they were covered by the phased blocks in this study. Overall, an average of 32% and 37% of such regulatory mutation candidates resided in the promoter and enhancer regions, respectively, and were assigned to either of the haplotypes (Supplementary

Fig. S11a). To examine the transcriptional consequences of the mutations in each haplotype, we performed long read RNA sequencing for six representative cancer specimens together with their normal counterparts (cases S1, S3, S6, S8, S10, and S20), for which high-quality RNA samples were available.

Based on the SNPs represented on the RNA-seq reads, the haplotype origin of their transcripts was identified. When an

**Fig. 3 Information on the chromosomal background of somatic mutations. a** Point mutations along with their haplotype information. The number of SNVs, multiple nucleotide variants and indels that are assigned to either of the haplotypes is shown for all examined 20 cases. The proportion of haplotype-resolved mutations is also represented in the graph. **b** The total number of SVs (left panel) and the ratio of SVs (right panel) are presented along with the corresponding haplotype information. **c** Cancer-related genes with phasing status. Point mutations and SVs in the 27 representative cancer-related genes are presented along with their corresponding phase information and block length. The legend and color key are provided in the margin. **d** Phasing of *EGFR* mutations in case S21. The IGV visualization of the PromethION reads (upper panel) and phase information (lower panel) are provided for two somatic mutations (R776S and L858R). Information on HP1 and HP2 are shown in red and blue, respectively. **e** Alignment patterns of long reads for each haplotype around a *PTEN* deletion in case S14 (upper panel). The sequencing depths of each haplotype around the *PTEN* deletion (lower panel). HP1: haplotype #1, HP2: haplotype #2, HPunk: haplotype unknown. **f** An example of multiple mutations of which the order of occurrence could be resolved by long read sequencing. Source data are provided as a Source Data file for **a** and **b**.

RNA-seq read harbored multiple SNPs, they were also used for comparison with the genomic haplotypes. The results of the comparison showed that the phasing was consistent, thereby confirming that the precise haplotype blocking is essentially evident in all examined cases (see Supplementary Fig. S11b for details). To assess the impact of A-to-I RNA editing for the haplotype misassignment of RNA sequences, we counted overlaps between the possible A-to-I RNA editing sites and A/G SNPs in the haplotype-resolved regions by RNA-seq (Supplementary Table S8). As a result, 0.22%–0.55% of A > G SNPs overlapped. The result indicates that the impact of RNA editing is insignificant.

From the RNA-seq data, an average of 718 genes (1185 SNPs) were differentially expressed among the haplotypes (Supplementary Fig. S11c). Among these, 133 tumor-specific haplotype-biased expressed genes (22 genes on average) were found to bare regulatory mutations in the promoter or enhancer regions in four cases; 115, 10, 7, and 1 genes with regulatory mutations were identified from cases S3 (a so-called "hypermutator" case), S8, S10, and S20, respectively (Supplementary Table S9). For the detected genes, their gene expressions may have appeared to be changed due to the detected regulatory mutations in each haplotype. In an example of the *CLN5* gene in case S10, a promoter mutation was found in HP2 (Fig. 4a). This mutation could strengthen binding potentials of the transcription factor (TF), such as SNAI2, which is known to be one of the master regulators of epithelial-mesenchymal transition (EMT)[24], by changing TF binding sites (TFBSs) (Supplementary Fig. S12 and Supplementary Table S10), which indicated that EMT-related factors might regulate the *CLN5* transcription instead of original TF components in the mutant promoter. Moreover, uneven gene expression was detected between HP1 and HP2 only in the tumor specimens (Fig. 4b). Precisely, six phased SNPs in the exonic region of this gene in the HP2 demonstrated substantially higher transcription levels than HP1 in the same tumor specimen (Supplementary Fig. S11d).

In an attempt to examine the epigenome statuses of these genes, we performed a DNA methylation analysis for all 20 cases. We used a bioinformatics program, Nanopolish, which is frequently employed for this purpose[25]. DNA methylation patterns were directly called from the raw electrograms of the PromethION WGS reads. Each read was assigned to either of the haplotypes using its harboring SNPs. An average of 32 haplotype-biased expressed genes were found with differentially methylated regions (DMRs) between HP1 and HP2 (Supplementary Table S9). We refocused on the *CLN5* gene of case S10 and identified a DMR among haplotypes in the 12-kb upstream of the gene (Fig. 4c). In HP2, where the promoter mutation was detected and the mRNA was strongly transcribed when compared with that of HP1, focal DNA hypomethylation was observed (right panel, Fig. 4c). In this region, various TFBSs, such as EMT-related factors, including SNAI2, could be bound so that the hypomethylation of this region might cooperate with the

promoter mutation and upregulate the downstream *CLN5* gene, accompanied by the EMT factors (Supplementary Fig. S12 and Supplementary Table S10). Collectively, these results demonstrate that the integrative analysis of the phased genomic, epigenomic and transcriptomic aberrations in cancer is possible and could provide essential information on the regulatory mutations that may exert inherently distinct effects on their residing haplotypes.

While identifying and characterizing the potential regulatory mutations, we noticed that the transcriptional aberrations occurring in a haplotype-biased manner did not always contain any obvious regulatory mutations. Obviously, the haplotype-biased expressions are not only caused by the genomic point mutations but also by other factors. We inspected the differentially transcribed SNPs between the haplotypes and found that 19% of the cases may be accounted for differences in the CN aberrations (haplotype-specific CN gains or loss) between haplotypes (Supplementary Figs. S11e and S11f). Nevertheless, for a substantial fraction of the remaining cases, each haplotype appeared to have significantly different epigenetic and transcriptomic landscapes. We could detect an average of 68 phase blocks differentially enriched with hypermethylated or hypomethylated regions between the haplotypes (Fig. 4d). In an example, a phased block (block#114040122) was identified baring five DMRs in case S10 while the HP2 of this block was broadly hypomethylated (Fig. 4e). Moreover, the transcription of the downstream *CDC16* gene was upregulated in HP2 (Supplementary Fig. S11g). Such aberrant methylation regions might have formed in a cancer genome inherently, where the consequential transcription also occurs. Each haploid genome may have a distinctive local context, which may have been developed due to unique molecular events in the past, and might provide unique molecular environments for the occurrence of further aberrant events leading to lung cancer in the future.

**Chromosomal features of the phased mutations and inferring the history of the lung cancer genomes**. Assuming a potentially haplotype-unique chromosomal background for each mutation, we wondered whether the mutations distribute truly randomly across the haplotypes. To address this issue, we tentatively focused on the genomic regions in which these mutations are particularly enriched and compared them to the overall mutation rate of the corresponding cancer genome (100-kb windows; Supplementary Fig. S13a). Among them, we further selected the genomic regions in which the mutations were statistically significantly biased to one of the haplotypes. We analyzed 834 of such mutation-enriched regions from the phased block regions of 15 cases. No mutation-enriched block was detected in the other five cases. The correlation between the number of mutation-enriched blocks and the tumor mutation burden (TMB) was evaluated (Spearman's $r = 0.78$, $p = 4.4e{-}5$, Supplementary Fig. S13b). The mutation-enriched blocks were undetected probably due to the small number of mutations. However, the

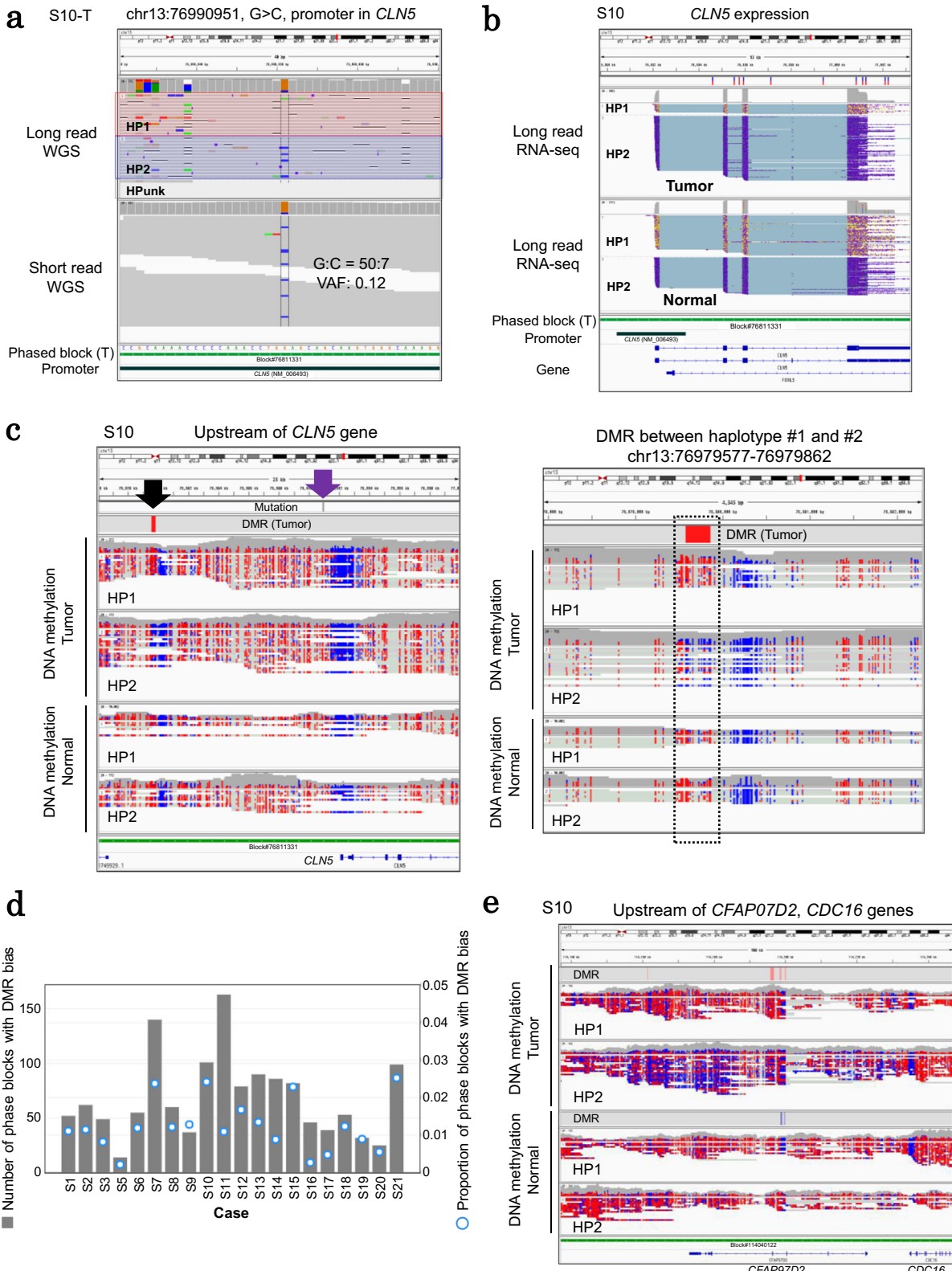

**a** S10-T    chr13:76990951, G>C, promoter in *CLN5*

**b** S10    *CLN5* expression

**c** S10    Upstream of *CLN5* gene

DMR between haplotype #1 and #2
chr13:76979577-76979862

**d**

**e** S10    Upstream of *CFAP07D2*, *CDC16* genes

chromosomal events accompanying the mutation-enriched blocks may have caused further mutational events, increasing TMB. Among them, 190 regions (1–75 region(s) per case) were identified to harbor more than 80% of the mutations preferentially in one of the haplotypes (designated as "haplotype-biased" regions; Fig. 5a).

Interestingly, the ratio of the haplotype-biased regions was found to be characteristically dependent on the cancer type or histological subtype (Fig. 5b). For example, mutations in these regions tended to be concentrated in one haplotype in typical lung adenocarcinomas. Moreover, they were randomly distributed to both haplotypes in other types of non-small cell lung

**Fig. 4 Haplotype-resolved transcriptional regulatory aberrations. a** An example of the haplotype-resolved regulatory aberrations of the *CLN5* gene in case S10. A promoter SNV was detected in the promoter region of *CLN5*, and this mutation was located on HP2. The visualization of long read (phase information) and short read (VAF of the mutation) sequencing is shown in IGV. **b** The expression pattern of *CLN5* in long read RNA-seq is demonstrated by IGV for tumor and normal datasets on the upper and lower panels, respectively. The reads were separated by each haplotype of block#76811331 where the promoter SNV (shown in **a**) was located. **c** DNA methylation statuses of the upstream region of the *CLN5* gene in case S10. A DMR between the haplotypes was observed (black arrow, left). The promoter mutation (shown in **a**) is represented by a purple arrow. A zoom-in visualization of the DMR is also presented (right). **d** The number of phased blocks with DMRs that were biasedly observed among the haplotypes. **e** An example of haplotype-resolved differential DNA methylation. DMRs between haplotypes and DNA methylation patterns on long reads are provided in IGV for both tumor and normal specimens for case S10. Source data are provided as a Source Data file for **d**.

cancers. This feature was particularly significant for the so-called "hypermutator phenotypes."

We further speculated on the local mutation patterns; particularly for the most relevant case S2, where 20 SNVs demonstrated significant local enrichment (adjusted $p = 5.0e-26$; Supplementary Fig. S13a). All 14 haplotype-resolved SNVs were assigned to HP1 (block#116042803, 2.8 Mb in length; Fig. 5c). These mutations had a predominant mutation pattern of a G > A substitution, which corresponded to the APOBEC mutational signature (SBS02; cosine similarity = 0.946; Fig. 5c)[26]. Moreover, the overall mutation signature of this case demonstrated a mixed pattern, which was distinct from the above local pattern (Fig. 5c). This observation indicated that this genomic region was exposed to a distinct mutational pressure from other regions. More generally, we examined whether similar base substitution patterns could be found in other regions. We found that the C > T/G > A and C > G/G > C substitutions tend to be the dominant mutation patterns in the mutation-enriched regions in a haplotype-biased manner (Fig. 5d). We also found that such a characteristic is unique to lung adenocarcinomas. In the other cancer types, the mutation patterns were rather similar to the overall genomic mutation patterns (Fig. 5d).

We similarly counted the SVs on the phased blocks and found six cases in which blocks contained over five SVs (Supplementary Fig. S13c). In these SV-concentrated phased blocks, most SVs were unevenly distributed to one haplotype (Fig. 5e, f). The six cases with the SV-concentrated phased blocks harbored a larger number of inversions and duplications ($p = 0.026$ and 0.0016, respectively; Wilcoxon rank-sum test, Supplementary Table S11). The mechanisms of SV occurrence might differ among SV types[10,27,28], and inversion and duplication types would tend to occur in a haplotype-specific manner compared with other SV types. These results also indicate that the difference in the characteristics between haplotypes is the reason for the occurrence of different mutational events between haplotypes.

**Chromothripsis-like event as a possible cause of haplotype-biased SNV/SV.** We further attempted to reveal the possible cause of the observed biased mutation distributions. We focused on case S20 in which both the mutations and SVs were particularly enriched in one haplotype. In this case, all SVs and SNVs in a large phased block (length: 3,746,985 bp) in chromosome 5 were detected in HP1 (Fig. 6a). We further inspected this haplotype block's structure and identified a cluster of interleaved SVs and CN oscillations (Fig. 6b). Both of these features are characteristics of a phenomenon called "chromothripsis"[29]. In chromothripsis, a large chromosomal region is damaged during an improper chromosome separation during the cell division. To repair the segmented chromosome, repair machinery is recruited to recover the chromosome but does so after leaving some "errors." The interleaved SVs are caused by the random joining of fragments after the chromosome shattering, and these fragments are detected as overlapping regions bridged by their breakpoints[29].

CN oscillation states indicate the regions with interspersed loss and retention of heterozygosity during chromothripsis[30].

Consistent with the previous knowledge of chromothripsis, the expected accumulation of C > T and C > G mutations, which is similar to that of the APOBEC signature, was observed at the same region of the SV clusters in case S20 (Fig. 6c, d and Supplementary Fig. S14). No other significant mutational signature (cosine similarity >0.7) was detected. Importantly, the APOBEC signature was reported to be frequently accompanied by chromothripsis. APOBEC3B has been suggested to access the single-stranded DNA in the intermediate process of chromothripsis and cause a phenomenon called "kataegis"[31,32]. In this case, we may be seeing traces of a chromosome-level genomic crush event, which has only happened once during the cancer history, rather than the accumulation of a series of individual mutational events.

To further clarify the characteristic features of these regions, we analyzed the DNA methylation status. We found that the HP1 region in which both SVs and SNVs were accumulated was left to a significantly hypomethylated status compared to both the same region of the HP2 and that of its matched-normal specimen (Fig. 6e and Supplementary Fig. S15a). To further investigate the transcriptional consequence of this hypomethylation, a phasing analysis similar to the aforementioned one was performed. The genes in the corresponding regions were generally transcribed evenly from both alleles in the matched-normal control genome, whereas almost all genes were transcribed solely from the HP1 of the tumor specimens (Fig. 6f and Supplementary Fig. S15). We could not further examine whether these hypomethylations and active gene expressions are either the cause or consequence of chromothripsis. However, at least, in this case, the other chromosome was never found to be at a similar condition.

**Characterization of the specimens having chromothripsis-like events.** Finally, to further investigate the possible large-scale chromosomal events, we analyzed all examined cases. We could detect similar traces of possible chromothripsis events from at least five specimens, although the affected sizes of the genomic regions varied. Interestingly, four of these cases were lung adenocarcinoma cases, and all of them bared *EGFR* mutations (L858R missense mutations and in-frame indels) as the driver mutation (Fig. 7a). Except for the lung adenocarcinoma cases, one particular case (case S16) was from a large cell neuroendocrine carcinoma. Although specimens of lung squamous cell carcinomas generally demonstrated a larger number of SVs than those of lung adenocarcinomas (Supplementary Fig. S4b), no obvious trace of a possible chromothripsis event was detected. A relationship between chromothripsis and the *EGFR* driver mutation in lung cancers has not been well-characterized. However, chromothripsis events have frequently been detected in *EGFR*-amplified glioblastoma specimens[33]. For lung cancers, lung adenocarcinomas baring mutations in *EGFR*, *KRAS*, *ERBB2*, and *MET* are reported to also host a higher number of "rearrangements"[34]. Particularly for lung adenocarcinomas, a

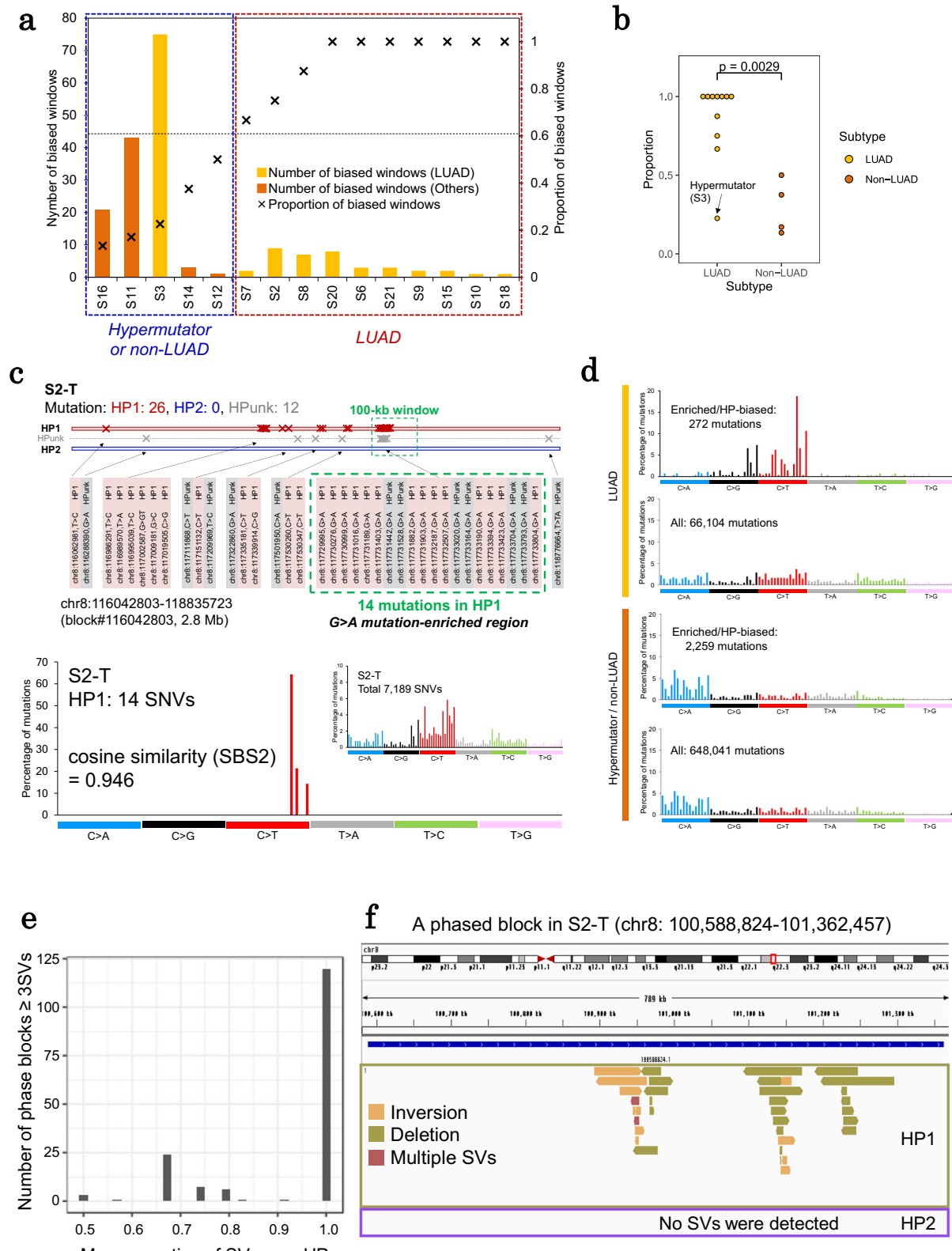

previous report showed that specimens with an *EGFR* driver mutation tend to have CN alterations and genomic instability, including whole-genome doubling, more frequently than other lung cancer species[35]. To further characterize chromothripsis-positive cases, we compared gene expression patterns with and without chromothripsis events in the *EGFR* mutation-positive cases. We found that pathways associated with inflammatory and

immune responses were upregulated in chromothripsis-negative cases (Supplementary Fig. S16). We suggested that this immune response might not be activated in chromothripsis-positive cases, and tumor cells with chromothripsis-like events would not be excluded. Immune escape ability in addition to the EGFR signaling activation would be a crucial factor for the progression of chromothriptic lung cancers.

**Fig. 5 Haplotype-biased mutation occurrences of lung cancer genomes. a** The number of phased blocks with an imbalance of point mutation occurrences between two haplotypes. The cases are ordered according to the proportion of the haplotype-biased regions of mutation occurrence. LUAD lung adenocarcinoma. **b** Comparison of the ratio of the haplotype-biased regions between LUAD and others. The *p* value was calculated by Wilcoxon rank sum test (two-sided, no multiple comparison adjustments). **c** A phased block with biased SNV occurrence as identified in case S2 (upper panel) and the mutational pattern of both the enriched and haplotype-biased mutations (lower panel). The mutation-enriched window is highlighted with a green-dashed line. **d** The mutational patterns of the enriched and haplotype-biased mutations compared with those of all mutations in LUAD and other hypermutator and non-LUAD cases. The number of mutations in each dataset is provided in the inset. **e** Association between the number of phased blocks containing three or more SVs and the maximum proportion of SVs occurring on one haplotype. **f** A phased block with biased SV occurrence in case S2. SV-supporting reads on this phased block were unevenly distributed to HP1. Source data are provided as a Source Data file for **a–e**.

To further elaborate on the relevance of chromothripsis, we examined the telomere length of each specimen. The telomere shortening in the tumor cells is reported to cause dysfunctional and unprotected telomeres, referred to as "telomere crisis." Telomere crisis yields end-to-end fused dicentric chromosomes and cause genome instability and chromothripsis[36]. We estimated the telomere length for each specimen using short read sequencing data. We found that all identified possible chromothripsis cases were presented with significant shortening of their telomere lengths (Fig. 7b). These results further support the idea that chromothripsis-like events occurred in our lung cancer specimens. Because the aberrant activation of the EGFR pathway might stimulate the telomerase activity[37], the relationship between chromothripsis and *EGFR* mutations in our specimens might be further elucidated by future analyses in this direction.

## Discussion

In this study, we demonstrated the phasing analysis of lung cancer genomes by the combinatory use of short and long read sequencing data. We generated a catalog of cancerous mutations of various types (such as SNVs and SVs) at a haplotype level. We found that the chromosomal backgrounds of the cancerous mutations are a rich resource toward the understanding of the biology of cancer genomes from various viewpoints. The transcriptomic or epigenomic features of cancer genomes could be better explained by also considering the chromosomal context of the genes of interest. We illustrated that, in some cases, the cancer genomes must have experienced large-scale aberrant events, which might have shaped the current form of cancer genomes. It is tempting to speculate that the large epigenomic and transcriptomic disordered regions identified in Fig. 4d, e may cause imperfect chromosomal aggregation or pairing at the cell division phase, and they may serve as a repertoire for future catastrophic events as indicated by Figs. 5–7.

Despite the initial success, we are aware that substantial technical developments are required to complete the long read cancer genome analysis. To facilitate the resolution of aberrant genomic events at a haplotype level, we reanalyzed our PromethON sequences using WhatsHap, which is one of the mapping-based phasing tools, assuming diploid genomes. We assigned somatic mutations into two haplotypes according to the haplotype-tagged PromethION reads. As cancer cells were heterogeneous because the mutations had been accumulating in tumor tissues during cancer evolution, an association between the mutations at a distance longer than reads by phasing analysis could not be guaranteed as actual phased events occurring in the same molecules. To actually "phase" multiple mutations at a distance longer than reads, which can be useful for analyzing cancer genomes, a single-cell analysis will be needed. We performed scDNA-seq for only two cases and were convinced that this combination analysis of long read phasing analysis and single-cell analysis would open further advanced fields for characterizing cancer genome evolutions. Further, we occasionally need to employ manual inspections to remove remaining errors and avoid considering

questionable regions. It is well-established that a considerable number of relevant events may occur in the aneuploidy regions of cancer genomes. The development of better algorithms that can systematically dissolve those regions should become a pressing issue. To this end, a de novo assembly-based method might be useful.

In experimental terms, technological developments have made rapid progress. For example, we obtained the much-needed SNP information using the Illumina short read data to complement the error-prone reads obtained from ONT sequencers. However, in the near future, we would perform a similar analysis solely based on long read data, once more accurate long read sequencing becomes possible. Easier and more cost-effective methods that would overwhelm even the upcoming PromethION Q20 platform will be developed.

In this study, we focused on characterizing the kataegis and chromothripsis regions in which SNVs and SVs were found to aberrantly accumulate in a haplotype. Chromothripsis has recently been highlighted as a critical event for tumorigenesis. Extrachromosomal DNAs deriving from lost segments due to chromothripsis can be unevenly distributed to daughter cells and cause amplification[38,39]. This amplification contributes to expressing drug resistance and developing tumor malignancy[40–42]. Further, several driver fusion oncogenes of lung adenocarcinoma are thought to be generated from chromothripsis[34]. These reports indicate that the understanding of the mechanisms and characteristics of chromothripsis could clarify potential targets for cancer therapy.

When combining our data with the allele-specific methylation status, we detected hypomethylation in a chromothripsis region. The relationship between hypomethylation and genomic instability has been suggested in previous studies[43–45] but has not been fully explored due to experimental limitations. Nanopore sequencing technology can detect nucleotides and the methylation status on each read simultaneously. In addition, longer haplotype information enables us to trace allele-specific events in the tumor. These advantages of the technology will provide us with in-depth insight into the tumor's genomic alterations. At this point, a novel challenge is emerging, eventually to reconstruct the cancer genome at chromosome levels[46]. We believe that the haplotype-resolved cancer genome sequencing will drive forward a new field of cancer genomics and be of invaluable assistance to researchers who use newer long read sequencing platforms.

## Methods

**Clinical specimens**. This study was approved by the Institutional Review Board of National Cancer Center, Japan and the Institutional Review Board of the University of Tokyo, Japan. Frozen surgical specimens from 20 patients were utilized[14]. All clinical specimens were obtained with the appropriate informed consent from the patients in National Cancer Center, Japan. The patient consent was obtained in a written form.

**WGS using PromethION**. The sequencing data used in this study were obtained from the previous report JGAS000065 (JGAD000252 and JGAD000253)[14]. We also obtained additional sequencing data of DNA samples to achieve sufficient

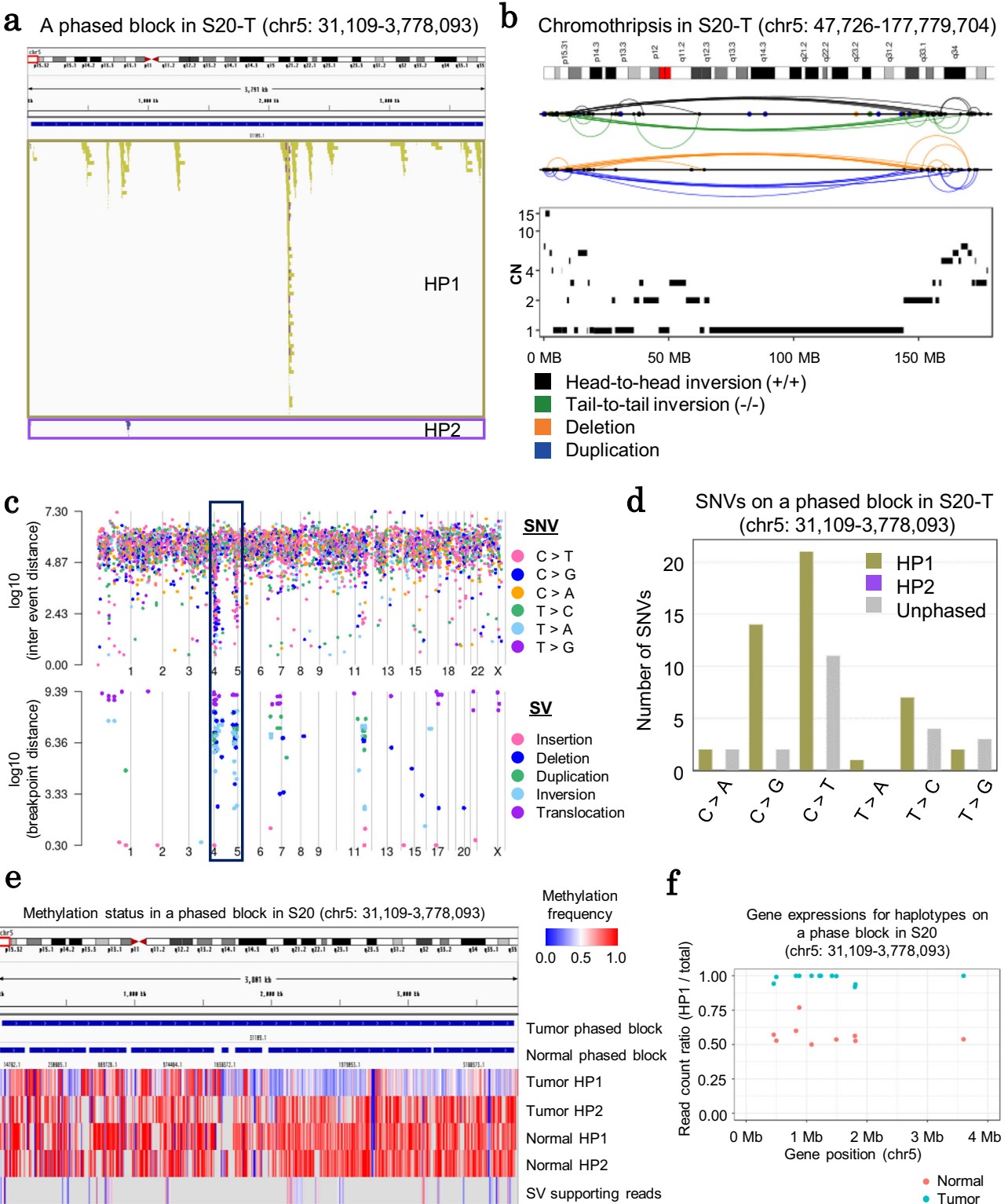

**Fig. 6 Chromothripsis on a haplotype level. a** An example of a phased block (chr5: 31,109–3,778,093) on which SVs have been found concentrated in case S20. Each color represents the haplotype of the reads supporting SVs. **b** An example of chromothripsis events in chromosome 5 as identified in case S20. The cluster of interleaved SVs is presented in the upper panel, whereas the CN patterns are presented in the lower panel. **c** Genome-wide information on the SNV and SV density of case S20. In the upper panel, the mutational patterns of the SNVs are represented in different colors and the potential kataegis is highlighted with arrows. Breakpoints for the SVs are shown for each respective color in the lower panel. **d** The number of SNVs corresponding to each haplotype in the phased block (chr5: 31,109–3,778,093). **e** Methylation status at the phase block located within the chromothripsis region of case S20. Normal and tumor panels demonstrate methylation frequencies calculated with reads assigned to each haplotype. The SV panel presents the frequencies calculated with SV-supporting reads detected by Nanomonsv. **f** Gene expressions for haplotypes at the phase block (chr5: 31,109–3,778,093) of case S20. Read ratios of HP1 and HP2 in normal and tumor tissues are shown with separate colors. Source data are provided as a Source Data file for **d**.

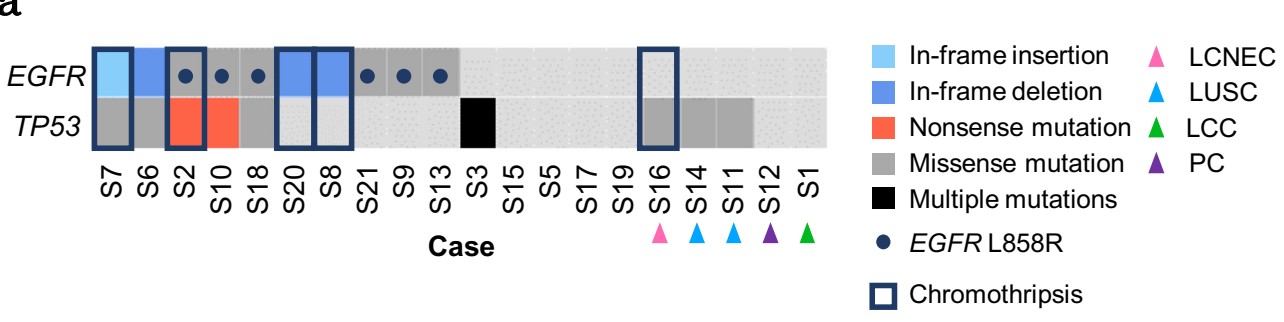

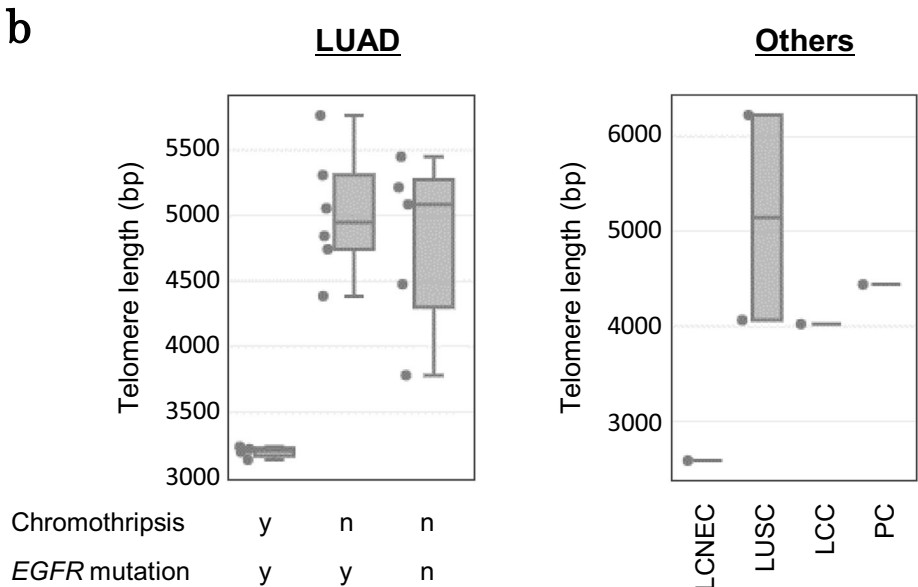

**Fig. 7 Characterization of specimens with chromothripsis. a** Relationship among mutation statuses and chromothripsis. The legend is provided in the margin. LUAD lung adenocarcinoma, LCNEC large cell neuroendocrine carcinoma, LUSC lung squamous cell carcinoma, LCC large cell carcinoma, PC pleomorphic carcinoma. **b** Comparison of the telomere length for each tumor type and each chromothripsis event. The ends of the box plots indicate lower and upper quartiles; center line, median; whiskers, maximum and minimum values, respectively. Source data are provided as a Source Data file for **b**.

sequencing depths, obtain appropriate phase information and conduct saturation analysis. Library preparation was conducted for 1D sequencing (SQK-LSK109 and SQK-LSK110, ONT), and sequencing analysis was performed according to the manufacturer's instructions.

For data acquisition using the PromethION Q20 platform, high molecular DNA samples of lung cancers obtained in the previous study were employed for the library preparation using the Q20 Early Access Kit (SQK-Q20EA, ONT) according to the manufacturer's instructions. Sequencing was performed by using PromethION with FLO-PRO111 flow cells (R10.3 version, ONT). Healthy human DNA sample HG002 (NA24385, Coriell Institute) was also sequenced for quality control.

**RNA-seq.** Total RNA was extracted from the frozen tissues using the TRIzol Reagent (Invitrogen). The RNA samples that yielded an RNA integrity number (RIN) > 6 in both tumor and normal specimens were used for constructing long and short read RNA-seq libraries. The RNA samples with RIN > 6 in only tumor samples were additionally used for short read RNA-seq analysis. The RIN values were measured by a Bioanalyzer (Agilent Technologies). Using the obtained RNA samples, full-length cDNA (FL-cDNA) amplicons were generated using the SMART-Seq v4 Ultra Low Input RNA Kit for Sequencing (634897, Takara Bio)[47].

For long read RNA-seq analysis, 1D sequencing libraries were prepared from FL-cDNA amplicons using the Ligation Sequencing Kit (SQK-LSK109, ONT). Sequencing was performed using PromethION (FLO-PRO002, ONT) according to the manufacturer's instructions.

For generating short read RNA-seq data, the library preparation was performed using a Nextera XT DNA Library Preparation Kit (FC-131-1096, Illumina) from the generated FL-cDNA amplicons. Sequencing with 150-bp paired ends was conducted using the NovaSeq 6000 (Illumina).

**scDNA-seq.** Individual single nuclei were extracted from frozen tumor tissues of cases S8 and S20 according to the manufacturer's demonstrated protocol "Isolation of Nuclei for Single Cell DNA Sequencing" (CG000167, Rev A, 10× Genomics). Library preparation of scDNA-seq was performed using Chromium Single Cell DNA Reagent Kits (10× Genomics) according to the instruction (CG000153, Rev C, 10× Genomics). The target cell recovery was set to 500 cells for each sample. Sequencing was conducted using NovaSeq 6000 with 150-bp paired-end reads (Illumina).

**Phasing analysis.** Phasing information on tumor and normal genomes was obtained in four steps: (1) short read WGS data (151 bp paired-end reads) of normal specimens were trimmed of their adapters and filtered using fastp (version 0.21.0)[48]; subsequently, their reads were mapped to the human reference genome hg38 using BWA-MEM (version 0.7.17)[49]; following their mapping, reads were sorted and PCR duplicates were marked using SAMtools (version 1.9)[50,51]. (2) The Genome Analysis Toolkit (GATK) HaplotypeCaller (version 4.1.7.0)[52,53] was used for detecting germline variants with a base quality and a variant quality score recalibration; after variant calling, variation sites with a "PASS" filter flag were extracted using BCFtools (version 1.9)[54], whereas SNPs were only used for further phasing analysis, (3) PromethION long read sequencing data from both tumor and normal genomes were mapped to hg38 using Minimap2 (version 2.17) with the "-ax map-ont" option[55]; the mapped reads were sorted and merged using SAMtools. (4) Phasing analysis was performed using the generated SNP Variant Call Format (VCF) files and the long read Binary Alignment Map (BAM) files of each sample by WhatsHap phase with the "--ignore-read-groups" option (version 1.0)[21], then SNP VCF files with phasing information were created; for visualization, a long read BAM file with tagged reads by haplotype was generated using the WhatsHap

haplotag. The detailed procedures of the cancer genome phasing are described in Supplementary Methods.

Switch/flip errors between the two datasets (tumor versus normal, current version versus Q20 version) were calculated using WhatsHap compare. The calculation was performed on each chromosome.

**Comparison of haplotype information with THC data**. Specimens of the THC dataset were collected from 418 healthy donors. WGS was conducted using the HiSeq X platform. Subsequently, sequenced data were mapped to the human reference genome hg38 using BWA-MEM (version 0.7.17). SNPs and short indels were called by GATK HaplotypeCaller (version 5.1), followed by joint calling. Joint calling was performed by the gVCFtyper program of the Sentieon package. Variant filtering was conducted using Variant Quality Score Recalibration by GATK. After filtering and quality control, the haplotype phasing was performed by BEAGLE (version 4.1)[56,57]. All singleton variants were omitted from the phasing.

Then, we compared the THC dataset with the phasing results obtained from our lung cancer datasets. We considered the two continuous phased heterozygous SNPs because the THC dataset provided one phased block by each chromosome, whereas our long read sequencing datasets provided several phased blocks in each chromosome. The latter is because, in the long read sequencing datasets, the assigned allele numbers were not meaningful when phase blocks differed. Considering this, we calculated the concordance between the THC dataset and our lung cancer datasets by focusing on the continuous phased heterozygous SNPs.

**Identification of somatic point mutations and their haplotype information**. Short read WGS data were mapped onto the human reference genome hg38 using BWA-MEM (version 0.7.17). Mapped reads were sorted and indexed by SAMtools (version 1.9), and duplicates were removed using Picard MarkDuplicates. Somatic SNVs and short indels were detected using GATK Mutect2 and FilterMutectCalls (version 4.1.3.0). The detected mutations were annotated using ANNOVAR (version 2018Apr16)[58]. For the regulatory mutation analysis, point mutations located on the promoters (±1.5 kb from RefSeq transcriptional start sites) and enhancers (H3K27ac/H3K4me1-marked regions defined in the previous studies[9,23]) were extracted. Mutations with the annotations of "exonic," "5UTR," or "3UTR" were excluded from the regulatory mutations.

For assigning haplotype information to somatic mutations, the SAMtools (version 1.12) mpileup function was used to extract base information from the haplotype-tagged PromethION reads in each mutation position. A haplotype tag (HP1 or HP2) was, respectively, assigned to each mutation when three or more mutant reads were assigned to the one haplotype and zero or one mutant read was assigned to the other.

**Identification of somatic SVs and their haplotype information**. Somatic SVs were detected from long read WGS data of tumor and normal genomes using Nanomonsv (version 0.1.2)[59] with default parameters. We filtered and classified the detected SVs using scripts provided on the Github page (https://github.com/friend1ws/nanomonsv). For the SV phasing, we selected and used SVs on chr1-22. Supporting reads for each SV were extracted and annotated using the haplotype tag in the WhatsHap results. Because WhatsHap calculates haplotypes only for primary alignment, the phased SNPs on the supplementary alignment reads supporting these SV were manually counted using the SAMtools (version 1.7) mpileup function. Then, we extracted the reads with ≥2 SNPs and a ratio of the number of SNPs for HP1 and HP2 ≥ 0.7 as a "phased read." We defined those that were supported with ≥3 phased reads and the ratio of the number of reads for HP1 and HP2 ≥ 0.7 as "phased SVs." Detailed information on the haplotype assignment of somatic SVs is described in Supplementary Methods and Supplementary Fig. S17.

**Estimation of clonal structures using short read WGS data**. PyClone-VI (version 0.1.1)[60] was used to estimate the number of clones and clonal architectures based on VAFs of somatic mutations from the results of GATK Mutect2. As recommended by the previous report[61], CN profiles and tumor purity were calculated by FACETS (version 0.6.2)[62] with the parameter cval = 400 of the procSample function as input of PyClone-VI. For male specimens, minor CNs of chromosome X were set to zero. When the NA value was calculated as tumor purity, half of the average of VAFs was set instead. Clone structures were inferred from the results of PyClone-IV using ClonEvol (version 0.99.11)[63].

**Clustering analysis and visualization of scDNA-seq data**. scDNA-seq data were analyzed using Cell Ranger DNA (v1.1, 10× Genomics) and Loupe scDNA Browser (v1.1, 10× Genomics). Noisy cells, which were defined by Cell Ranger DNA, were discarded before further analyses. The CN profiles of 20-kb windows were extracted from the results of Cell Ranger DNA. Clustering analysis of cells was performed using the Manhattan distance via the complete linkage method. Heatmap visualization was performed using the R package pheatmap (version 1.0.12).

Reads with the point mutations were extracted from the BAM file of Cell Ranger DNA using SAMtools mpileup with the parameters as follows: the minimum base quality value was set to 20; the minimum mapping quality value was set to 60; the secondary, supplementary, and duplicate reads were discarded.

To extract scDNA-seq reads covering the SV junctions, primary soft-clipped reads and their supplementary alignments, which covered positions within ±100 bp of both junctions, were extracted from the BAM file of Cell Ranger DNA using SAMtools. For extracting junction reads of deletion, duplication and inversion types, we also checked strand information. Mutant cells were defined according to the cell barcode tag of the SV junction reads.

**DNA methylation analysis**. CpG methylation sites were called from fast5 data, and we calculated their methylation frequency using nanopolish (version 0.13.2)[25,64] under the settings as shown in Supplementary Methods. Using the haplotype-tagged PromethION reads from "WhatsHap haplotag," we classified the reads into two haplotypes and calculated the methylation frequencies in each haplotype category using the given script in nanopolish. We converted the BAM file by nanopore-methylation-utilities[65] for visualization in the Integrative Genomics Viewer (IGV)[66] (version 2.8.9 and 2.11.9) via the bisulfite mode using nanopolish results. The methylation frequency for the region covered with ≥3 reads was visualized as a heatmap in IGV. DMRs between the haplotypes were detected by metilene (v.0.2-8)[67]. We defined the blocks satisfying the following conditions as "phase blocks with DMR bias." (i) The maximum number of DMRs in the same direction being ≥3. (ii) The ratio of (i) for total DMRs being ≥0.7. (iii) The number of (i) per megabase (Mb) being ≥3. To accurately count (i), we excluded the DMRs that overlapped with other DMRs in the matched-normal specimen over 20% of the region. Detailed information on the haplotype-specific methylation analysis is presented in Supplementary Methods and Supplementary Fig. S17.

**Transcriptome analysis**. For long read RNA-seq analysis, FL-cDNA-seq data (1D reads) from PromethION were aligned to the human reference genome hg38 using Minimap2 (version 2.2.17) with the "-ax splice" option. Filtering for low-quality reads and eliminating pseudogene mapping[68] were performed using the parameters as described the Supplementary Methods. Similar to the SV phasing procedure, we counted the phased SNPs on the reads using the SAMtools (version 1.7) mpileup function. The reads with ≥2 SNPs and a ratio of the number of SNPs for HP1 and HP2 ≥ 0.7 were defined as "phased reads."

For short read RNA-seq analysis, paired-end reads obtained from the NovaSeq 6000 (Illumina) were aligned to the human reference genome hg38 using STAR (version 2.7.5c). Sequencing depths and tag densities (PPM, parts per million reads) of each haplotype were extracted using SAMtools (version 1.12) mpileup for haplotype-tagged SNP positions of exonic regions.

For gene expression analysis, short read RNA-seq data were mapped to the reference genome hg38 using STAR (version 2.7.5c) after conducting adapter trimming by fastp (version 0.23.2)[48] and removing ribosomal RNA sequences by Bowtie 2 (version 2.3.4.3)[69]. Gene expression levels (count and RPKM, reads per kilobase of exon per million mapped reads) were calculated using featureCounts (version 2.0.2)[70]. Differentially expressed genes (DEGs) were extracted using DESeq2 (version 1.32.0)[71]. Each parameter of DESeq2 analyses was provided in the corresponding figure legend. Gene enrichment analysis of the DEGs was conducted using Metascape (release 3.5)[72].

**Identification of potential TFBSs in regulatory regions**. To identify TFBS candidates potentially bound to mutant/wild-type sequences of the *CLN5* promoter, we scanned TFBSs registered at the JASPAR database (JASPAR2020, version 0.99.10)[73] at ±10-bp regions from the mutation (chr13:76990951, G > C) using the searchSeq function (min.score = 80%) of TFBSTools (version 1.30.0)[74]. The parameter of JASPAR2020 was set as follows—"species": 9606, "all_versions": TRUE, "collection": CORE, "tax_group": vertebrates, and "matrixtype": PWM. The candidates were further extracted with a *p* value (sampling) < 0.001 on either mutant or wild-type sequences.

For the DMR upstream of the *CLN5* gene (chr13:76979577-76979862), putative TFBSs were also searched similarly. We also checked "JASPAR Transcription Factors" at the UCSC Genome Browser (GRCh38/hg38)[75] to represent the distribution of TFBSs.

**Detection of chromothripsis**. Somatic CN variants (CNVs) were detected using short read WGS datasets through Control-FREEC (version 11.6)[76,77] with the following parameters: ploidy = 2, window = 5000, and minimalSubclonePresence = 20. We clustered the interleaved SVs using bedtools (version 2.27.1) and counted the number of oscillating CN states in the cluster.

After considering the previous reports[29,78], we defined those satisfying the following conditions as "chromothripsis" regions. (i) The number of SVs in the cluster being ≥10. (ii) The number of oscillating CN segments in the cluster between two CN states being ≥4 or among three CN states being ≥6. (iii) The number of SVs per Mb in the cluster being ≥0.2. (iv) The number of CNV states per Mb in the cluster being ≥0.2. SV cluster and CN states in the chromothripsis region were visualized using ShatterSeek (version 0.4)[29].

**Estimation of telomere length**. We used short read WGS data for estimating the telomere length using Telomerecat (version 3.4.0) with default parameters[79].

**Reporting summary**. Further information on research design is available in the Nature Research Reporting Summary linked to this article.

## Data availability

The data of clinical specimens including long read WGS, scDNA-seq and RNA-seq data generated in this study have been deposited in the Japanese Genotype-Phenotype Archive (JGA, http://trace.ddbj.nig.ac.jp/jga), which is hosted by the National Bioscience Database Center (NBDC) and the DDBJ under the accession number JGAS000349. These data are available under restricted access due to ethical restriction. Details of the procedure and the restriction for the data access are described in the home page of the JGA database [https://humandbs.biosciencedbc.jp/en/data-use]. The sequencing data of HG002 was deposited to DDBJ under the accession number DRA012759. The sequencing data including short and long read WGS obtained in the previous study of ours JGAS000065 (JGAD000252 and JGAD000253)[14] was also used in this study, which are available under restricted access due to ethical restriction. The human reference genome hg38 was downloaded from the UCSC Genome Browser (https://hgdownload.soe.ucsc.edu/downloads.html). Source data of the figures are provided with this paper.

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

## Acknowledgements

We thank K. Imamura, K. Abe, M. Tsubaki, Y. Kuze, E. Ishikawa and S. Minamiguchi for their technical assistance. We are grateful to the National Cancer Center Biobank. This work was supported by the JSPS KAKENHI Grant Number 20H00545 (to T.K.), 21J13203 (to Y. Sakamoto) and 16H06279 (PAGS) (to Y. Suzuki). This work was supported by the Japan Agency for Medical Research and Development (AMED) P-CREATE grant number JP21cm0106582 (to A.S.) and 21cm0106577 (to T.K.). The supercomputing resource was provided by the Human Genome Center, the University of Tokyo (http://sc.hgc.jp/shirokane.html).

## Author contributions

M.S., Y. Suzuki and A.S. designed the study. Y. Sakamoto, S.M., M.O., Y. Shiraishi and A.S. contributed to the analysis of the sequencing data. A.K. conducted sequencing experiments. Y.K. and K.T. conducted the phasing analysis of the Japanese cohort. T.K. coordinated the specimens. S.N., Y. Shiraishi, T.K and Y. Suzuki interpreted the findings and supervised the study. Y. Sakamoto, S.M., M.O., M.S., Y. Suzuki and A.S. wrote the manuscript. All authors have approved the final version of the manuscript.

## Competing interests

The authors declare no competing interests.
