## [Peer Review file · Nature Communications]

REVIEWER COMMENTS

Reviewer #1 (Remarks to the Author): Expert in lung cancer genomics

The manuscript by Dr. Yoshitaka Sakamoto et al. on " Phasing analysis of lung cancer genomes using a long-read sequencer ", reported the phasing analysis of lung cancer genomes by the combinatory use of short read and long read sequencing technologies. This work provided the haplotype information of mutations identified from previous patient data through a combinatory use of short read and long read sequencing technologies. The results indicated that the read length provides a greater contribution to the haplotype phasing than the sequencing depth. They also found the transcriptional aberrations occurred in a haplotype-biased manner, and the focal DNA methylation status was consistent with the haplotype-biased expressions. This study promotes the development of haplotype-resolved cancer genome sequencing, which might shed new light on better understanding of tumor biology.

Major comments:

1. Regarding the occurrence order of multiple mutations, phylogenetic analysis could be run as a validation. And single-cell DNA sequencing data would be a better option to further validate.
2. Regarding the integrative analysis of the phased genomic, epigenomic and transcriptomic aberration, how do the authors interpret the causational relation between the regulatory mutations, DNA methylation and transcription?
3. "No mutation-enriched block was detected in the other four cases, perhaps due to the fact that these cases originally harbored a small number of mutations." TMB of each sample can be analyzed to check if there is any correlation between them across all samples.
4. The SV-concentrated phased blocks were only found in 6/20 cases. Do these 6 samples have very different features compared to the other 14 samples?
5. An experimental validation of the activation of the pathway described would have been a great addition.

Minor comments:

1. Line 59-60, please spell "SNV" and "SVs" at the first appearance.
2. Line 137, what was the average length of short-read sequencing reads?
3. The figures should be modified by using uniform font size and color etc. Please carefully check the grammars throughout the manuscript.

Reviewer #2 (Remarks to the Author): Expert in genomics, bioinformatics, long-read sequencing and phasing

The authors performed genome-wide haplotype-specific SNV, SV, methylation, and transcriptome analysis in 21 cancer samples and their normal counterpart using Nanopore sequencing and Illumina sequencing. The results were obtained using a number of existing algorithms and some in-house script. Studying cancer genome, epigenome, and transcriptome in a haplotype-specific manner is an important topic and technically feasible with the latest sequencing technologies and bioinformatic algorithms. However, I have following concerns regarding the manuscript writing, methods and results:

1. Most statements in Introduction are lack of support from either literature or experiments of this manuscript. For example, there is no literature citation in statements "A number of important drug targets and biomarkers have been identified, such as EGFR and BRAF." in page 4. The Introduction section should be completely revised by adding citations to every scientific conclusion.

2. Software parameters are not included in the manuscript, which are critical to the conclusions of the study. This includes but not limited to: 1) how to process reads mapped to multiple locations during reads alignment; 2) WhatsHap assumes that the number of haplotypes are fixed. What is the number of haplotypes used in this study?. These might affect the observations and conclusions. For example, the authors observed that some high depth regions have shorter phased blocks. If a region has multiple copies in the genome, some reads from other non-identical copies might be mapped to this region, this will make the data looks like having more than two haplotypes. If the number of haplotypes in WhatsHap is set to 2, the algorithm might be confused and reports shorter phased blocks.

3. The authors claim that they obtained about 0.2Mb ~ 8.5 Mb long phased SNV blocks using ~20kb long nanopore reads. These numbers are reasonable for phasing germline SNVs. However, I am not convinced that somatic SNVs with distance longer than reads can be unambiguously phased according to the described methods. The key difference between phasing germline SNVs and phasing somatic SNVs is that the number of haplotypes is known for phasing germline SNVs but unknown for phasing somatic SNVs.

When the number of haplotypes is known, for example in the case of phasing germline SNVs in a diploid genome, assuming that the reads mapping is unique and perfectly correct, a phased block can be

unambiguously extended as long as there are a couple reads spanning two heterozygous SNVs. Thus, it is feasible to obtain phased blocks much longer than reads (Figure C1).

When the number of haplotypes is unknown, for example in the case of phasing somatic SNVs in cancer, microbiome, or highly mutable virus, a phased block cannot be unambiguously extended (Figure C2).

The methods used in this manuscript phase germline SNVs first and then assign somatic SNVs to the inferred haplotype by looking at the linkage between somatic SNVs and their neighboring germline SNVs. This can NOT phase multiple somatic SNVs with distance longer than reads as explained in Figure C1 and Figure C2 since the samples might have more than two haplotypes induced by somatic SNVs. The authors should either phase distant somatic SNVs using better algorithms/technologies or acknowledge that somatic SNVs with distance longer than reads actually can NOT be phased unambiguously. The current writing in Results should be also revised because it gives readers the incorrect impression that somatic SNVs with distance longer than reads can be phased.

Phasing SVs has the same problem.

4. The authors used Nanopolish to detect DNA methylation. Although a couple of other tools can report haplotype-specific methylation, Nanopolish does NOT do it. The authors should clarify how they obtain haplotype-specific methylation or acknowledge that the methylation profiles they obtained are not haplotype-specific.

5. The haplotyping results using RNA-seq reads harboring multiple SNPs were compared to haplotype inference using gDNA sequencing data. This is plausible in general, but the impact of RNA editing, which might affect the results, is not discussed.

The authors performed genome-wide haplotype-specific SNV, SV, methylation, and transcriptome analysis in 21 cancer samples and their normal counterpart using Nanopore sequencing and Illumina sequencing. The results were obtained using a number of existing algorithms and some in-house script. Studying cancer genome, epigenome, and transcriptome in a haplotype-specific manner is an important topic and technically feasible with the latest sequencing technologies and bioinformatic algorithms. However, I have following concerns regarding the manuscript writing, methods and results:

1. Most statements in Introduction are lack of support from either literature or experiments of this manuscript. For example, there is no literature citation in statements “A number of important drug targets and biomarkers have been identified, such as EGFR and BRAF.” in page 4. The Introduction section should be completely revised by adding citations to every scientific conclusion.

2. Software parameters are not included in the manuscript, which are critical to the conclusions of the study. This includes but not limited to: 1) how to process reads mapped to multiple locations during reads alignment; 2) WhatsHap assumes that the number of haplotypes are fixed. What is the number of haplotypes used in this study?. These might affect the observations and conclusions. For example, the authors observed that some high depth regions have shorter phased blocks. If a region has multiple copies in the genome, some reads from other non-identical copies might be mapped to this region, this will make the data looks like having more than two haplotypes. If the number of haplotypes in WhatsHap is set to 2, the algorithm might be confused and reports shorter phased blocks.

3. The authors claim that they obtained about 0.2Mb ~ 8.5 Mb long phased SNV blocks using ~20kb long nanopore reads. These numbers are reasonable for phasing germline SNVs. However, I am not convinced that somatic SNVs with distance longer than reads can be unambiguously phased according to the described methods. The key difference between phasing germline SNVs and phasing somatic SNVs is that the number of haplotypes is known for phasing germline SNVs but unknown for phasing somatic SNVs.

When the number of haplotypes is known, for example in the case of phasing germline SNVs in a diploid genome, assuming that the reads mapping is unique and perfectly correct, a phased block can be unambiguously extended as long as there are a couple reads spanning two heterozygous SNVs. Thus, it is feasible to obtain phased blocks much longer than reads (Figure C1).

Figure C1. Haplotype inference with fixed number of haplotypes (diploid)

When the number of haplotypes is unknown, for example in the case of phasing somatic SNVs in cancer, microbiome, or highly mutable virus, a phased block cannot be unambiguously extended (Figure C2).

Figure C2. Haplotype inference with unknown number of haplotypes

The methods used in this manuscript phase germline SNVs first and then assign somatic SNVs to the inferred haplotype by looking at the linkage between somatic SNVs and their neighboring germline SNVs. This can NOT phase multiple somatic SNVs with distance longer than reads as explained in Figure C1 and Figure C2 since the samples might have more than two haplotypes induced by somatic SNVs. The authors should either phase distant somatic SNVs using better algorithms/technologies or acknowledge that somatic SNVs with distance longer than reads actually can NOT be phased unambiguously. The current writing in Results should be also

revised because it gives readers the incorrect impression that somatic SNVs with distance longer than reads can be phased.

Phasing SVs has the same problem.

4. The authors used Nanopolish to detect DNA methylation. Although a couple of other tools can report haplotype-specific methylation, Nanopolish does NOT do it. The authors should clarify how they obtain haplotype-specific methylation or acknowledge that the methylation profiles they obtained are not haplotype-specific.

5. The haplotyping results using RNA-seq reads harboring multiple SNPs were compared to haplotype inference using gDNA sequencing data. This is plausible in general, but the impact of RNA editing, which might affect the results, is not discussed.

Point-by-point responses to Reviewer #1:

The manuscript by Dr. Yoshitaka Sakamoto et al. on "Phasing analysis of lung cancer genomes using a long-read sequencer ", reported the phasing analysis of lung cancer genomes by the combinatory use of short read and long read sequencing technologies. This work provided the haplotype information of mutations identified from previous patient data through a combinatory use of short read and long read sequencing technologies. The results indicated that the read length provides a greater contribution to the haplotype phasing than the sequencing depth. They also found the transcriptional aberrations occurred in a haplotype-biased manner, and the focal DNA methylation status was consistent with the haplotype-biased expressions. This study promotes the development of haplotype-resolved cancer genome sequencing, which might shed new light on better understanding of tumor biology.

Major comments:

1-1. Regarding the occurrence order of multiple mutations, phylogenetic analysis could be run as a validation.

We thank the reviewer for this constructive comment. To validate the occurrence order of multiple mutations or characterize the obtained data from a different viewpoint, we have performed the phylogenetic analysis. We estimated the clone architectures of each case using PyClone-VI based on variant allele frequencies (VAFs). For the initial PyClone analysis, we used the short read WGS data exploiting its deeper sequencing depth than that of long read data. As the results of the PyClone analysis, we could detect multiple cancer cell clones (mutation clusters) in 11 cases (**Supplementary Fig. S9b**). We also mapped mutation pairs that were directly supported by the individual long reads to these inferred clone structures. We successfully validated 44 mutation pairs that could be mapped to the different clones, and hence could explain the correct clonal structure, at least, in five cases. Despite the general consistency, we found that the order of the mutation occurrence indicated by the long read information in one mutation pair was inconsistent with that by the PyClone-VI analysis. We further manually scrutinized this case and now are considering that this erroneous detection may have been derived from the fact that PyClone-VI could not properly handle the intensive chromosomal copy number (CN) aberration at the corresponding position (at

Chromosome 5 in case S20). We also found that, generally, inferring clonal structures in such regions would be inherently error-prone in PyClone analysis. The list of these consistent cases and the inconsistent case is presented in **Supplementary Table S6**. A successful example in two mutations in the *PDGFD* intron in case S20 is also presented in **Supplementary Figures S9e** and **S9f**. We have also added these results and related discussions to the revised manuscript (**Lines 307–311**).

Further, among the observed mutation pairs, we particularly wished to verify the relationship between the *PTEN* deletion and *PIK3CA* mutation in case S14, which was originally described in **Figure 3**. In this case, the PyClone analysis inferred two models of clone structures (branched and linear), each of which comprised three clones (**Supplementary Fig. S9c**). Among the two models inferred by ClonEvol, we found the linear model should be realistic because the order of two mutation pairs that were assigned to the clonal structures from C-2 to C-3 was directly supported by the long reads (**Supplementary Fig. S9d**). In the linear model, the *PIK3CA* mutation was assigned to the clone occurring in the most minor fraction, whereas the *PTEN* deletion should have occurred in the most major fraction, where the *TP53* mutation was also included. This model was consistent with the one we discussed in the original manuscript.

In this line of the analysis, too, despite the general success of the validation, we found that PyClone still collectively mapped a large fraction of the individual mutation pairs to the same clones. The lack of resolution may be derived from the insufficient sequencing depths of the short read data to resolve minor clones with low VAFs. However, we envision that long read sequencing would more sensitively detect the occurrence of mutations adjacent to each other if a similar sequencing depth is available in the future. For this purpose, a further improved version of PyClone, which can handle long read sequence data as well, should be also developed. We have added this discussion to the revised manuscript (**Lines 314–317**).

1-2. And single-cell DNA sequencing data would be a better option to further validate.

Accordingly, we have also performed single-cell DNA sequencing analysis for two representative cases, cases S8 and S20. In these cases, we describe the possible

“chromothripsis-like events” occurred. We extracted nuclei from frozen tissues and conducted single-nucleus DNA sequencing using Chromium Single Cell CNV solution (10x Genomics; note that this kit production has been already discontinued). (**Supplementary Table S7**). We performed clustering analysis using the obtained single-cell CN profiles and identified several cell clusters (**Supplementary Fig. S10a**).

First, we mapped point mutations to individual cells by examining whether the mutant reads were found in the single-cell DNA sequencing data of each cell. As shown in **Supplementary Figure S10b**, a substantial number of mutations were represented in the single-cell DNA sequencing reads. Of note, mutations, especially “minor” mutations (cluster C-2), were not always represented (**Supplementary Figs. S10c** and **S10d**). This is derived from the fact that, given the limited sequencing coverage and due to the possible “allele-drop” (experimental loss from the just two of the chromosomal DNA molecules existing within a single cell), a particular number of cells should be collectively analyzed to identify and characterize point mutations and SVs.

Nevertheless, we could investigate the cases of the mutation pairs for which their occurrence orders were directly resolved by the long reads. Two and thirteen mutation pairs were also covered in individual cells in cases S8 and S20, respectively. Two of the examples are shown in **Supplementary Figures S10e** and **S10f**, and the examples are described in the legend. We could further map SVs detected from long reads to single-cell DNA sequencing reads (**Supplementary Figs. S10g–i**).

In general, from this series of extensive analyses, we found that the combination analysis of long read phasing analysis and single-cell DNA sequencing would open further advanced fields for characterizing cancer genome evolution. (In fact, we have requested 10x Genomics R&D to consider reproducing this kit, having observed these results). In fact, we consider it is possible to deepen the analysis even with the current dataset, but please allow us to refrain from going further. We are afraid that such an analysis should not be within the scope of this particular manuscript. These results have been added to the revised manuscript (**Lines 311–313**; also see the legend of **Supplementary Figure S10**). The related discussion has also been enriched in the **Discussion** section (**Lines 551–555**).

2. Regarding the integrative analysis of the phased genomic, epigenomic and

transcriptomic aberration, how do the authors interpret the causal relation between the regulatory mutations, DNA methylation and transcription?

We appreciate this thoughtful comment from the reviewer. To resolve the haplotype-aware causal relation among a series of possible aberrant events occurring at several omic layers, we further inspected the possible transcription factor binding sites (TFBSs) of the regulatory regions where mutations resided or differential methylation occurred. The results are summarized in **Supplementary Figure S12** and **Supplementary Tables S9** and **S10**. We have also included the related discussion to the revised manuscript (**Lines 345–349, 352–357, 366–368, and 372–376**).

To demonstrate the results, we presented the case of the *CLN5* gene regarding its haplotype-aware expression aberration in association with its genomic mutation (regulatory mutation) or epigenomic aberrations (differential methylation), which were also observed in a haplotype-aware manner. The case of the *CLN5* gene in case S10, which was the originally exemplified case in **Figure 4**, is, again, exemplified in **Supplementary Figure S12**.

In this particular case, a detected genomic mutation in its promoter region resulted in the weakened binding scores of several transcription factors (TFs). The TFBSs included those of ZBTB6 and ZNF341. Moreover, the binding site of SNAI2 was newly created (the binding score was strengthened) by this mutation. *SNAI2* gene is known to be one of the master regulators of epithelial-mesenchymal transition (EMT) (Cobaleda C et al. 2007 *Annu Rev Genet*). We confirmed the significant level of gene expression of SNAI2 (10.0 rpkm) in case S10 by our RNA-seq data. It is possible that some EMT-related factors, including SNAI2, should come to regulate the *CLN5* transcription instead of the original TF components in the mutant promoter. This discussion has been added to the revised manuscript (**Lines 352–357, Supplementary Fig. S12a**). We also examined potential TFBSs of the HP2-specific hypomethylated region in the 12-kb upstream of the gene (**Supplementary Fig. S12b**). Several TFBSs of EMT-related factors, including SNAI2 itself, were also detected in this region. We further searched public cell line data (Suzuki et al. 2018 NAR) for the open chromatin/enhancer regions and found that these two regions both form relevant open chromatin structures (**Supplementary Fig. S12c**). Based on these data, we consider that the detected promoter mutation may have “caused” the generation of the upstream

hypomethylated region and the promoter of the *CLN5* gene to interact more intensively via the binding of SNAI2. These events may cooperatively realize the “subsequent” upregulation of the *CLN5* expression. This discussion has also been added in the revised manuscript (**Lines 372–376**). We also included the details of these analyses in the legend of **Supplementary Figure S12**.

3. “No mutation-enriched block was detected in the other four cases, perhaps due to the fact that these cases originally harbored a small number of mutations.” TMB of each sample can be analyzed to check if there is any correlation between them across all samples.

Accordingly, we have assessed the correlation between the number of mutation-enriched blocks and tumor mutation burden (TMB) for each case. We found a strong positive correlation between them (Spearman’s $r = 0.78$, $p = 4.4e-5$, **Supplementary Fig. S13b**). We have added the results in the revised manuscript (**Lines 410–412**). The mutation-enriched blocks were probably undetected simply because the small number of mutations. However, it is also possible that the chromosomal events, such as chromothripsis-like events, may have caused further mutational events, resulting in the increased TMB. We have also included this discussion in the revised manuscript (**Lines 412–415**).

In addition, we are sorry for the typo in the number of cases in the original manuscript. There are mutation-enriched regions in 15 cases (the original version of **Figure 5a** represents the precise number) and no mutation-enriched block was detected in the five (not four) cases (numbers in the figures were correct). We have also corrected these numbers in the revised manuscript.

4. The SV-concentrated phased blocks were only found in 6/20 cases. Do these 6 samples have very different features compared to the other 14 samples?

To address this issue, we compared available pieces of clinical/pathological and

mutation information for the specimens. We examined whether there is any different feature between the six cases and the other fourteen cases. We found no statistically significant difference in the patient backgrounds or the pathological information between them (**Supplementary Table S11**). Nevertheless, we found that the six cases with the SV-concentrated phased blocks harbored numerous inversions and duplications ($p = 0.026$ and 0.0016 , respectively; Wilcoxon rank-sum test) than the other cases. The mechanisms of SV occurrence are supposed to differ among various SV types (Yang L et al. 2013 *Cell*, Wang WJ et al. 2020 *Epigenetics Chromatin*, Li Y et al. 2020 *Nature*). The mechanism causing the inversion and duplication types may be associated with the concentrated SV distribution pattern having a haplotype-bias. We have added this discussion to the revised manuscript (**Lines 443–447**).

5. An experimental validation of the activation of the pathway described would have been a great addition.

To address this comment, we performed RNA-seq for additional five cases (S2, S7, S11, S14, and S16, for which the quality of RNA satisfied the criterion of RIN > 6 in tumor specimens). The additional cases used for analysis include the cases we had presented to indicate characteristic mutations in the original manuscript. In this revision, we integrated all RNA-seq datasets from 11 cases to characterize pathway activation as a whole dataset (**Supplementary Fig. S8a**).

First, we examined the overall gene expression profiles. As shown in **Supplementary Figures S8a** and **S8b**, the gene expression patterns could be separated into LUAD and other lung cancer species using the indicated genes. The responsible genes for the clustering included *NAPSA* and surfactant proteins. Nevertheless, we also found that, while the gene expression patterns were generally different between cancer species or individuals, some parts of the gene expression patterns in particular pathways are resembled occasionally, perhaps reflecting the mutation statuses of the associated genes.

Firstly, we closely inspected the gene expression profile of S14, where a large deletion of the *PTEN* gene and the point mutation of *PIK3CA* were detected (as originally shown in **Figure 3**) and compared it with those of other cases. Particularly, we

examined the expression inductions of the genes associated with the PTEN/PI3K-AKT pathway, possibly occurring as the transcriptional addition of the pathway activation. Firstly, we could confirm that the *PIK3CA* mutation (E545K) was directly represented on the RNA-seq tag sequences, showing that this mutant allele is actively transcribed. The expression level of the mutant allele was even higher than the wild-type allele, suggesting that cancer cells, in this case, should be addicted to the *PIK3CA* mutation (**Supplementary Figs. S8c and S8d**). Further, we evaluated the differential expression between cases S14 and S11 which were both derived from the same squamous cell carcinoma. Case S11 is a *PIK3CA* mutation-negative case and showed differential expression patterns with case S14 potentially regarding the PI3K-AKT signaling activation that might specifically occur in case S14 (**Supplementary Fig. S8e**). We further inspected the expression levels of cell cycle genes because one of the pivotal phenotypic functions of this pathway is to facilitate cellular proliferation. As expected, we detected that the genes associated with cancer cell proliferation were highly expressed in case S14, which were compatible with those of other cases (**Supplementary Fig. S8f**). Interestingly, the gene expression level of *CCND2* was particularly high, whereas that of *CCND1* was relatively low in case S14. Analyses of these molecules will reveal details of aberrant signals in which cancer cells would be addicted in a case-specific manner.

In general, these results showed that the aberrant activation of the PTEN/PI3K-AKT pathway occurs in case S14, at least to no less relevant levels with other lung cancers, and the detected *PIK3CA* mutation in the context of the *PTEN* deletion should be responsible for this aberrant activation. The results of these analyses have been included in the revised manuscript (**Lines 292–294**).

In addition to characterizing transcriptomic features of the PTEN and PI3K-AKT pathway, we performed the similar pathway analysis to characterize and validate transcriptome features of the cases in which the chromothripsis-like events were observed. To characterize the chromothripsis-positive cases in addition to the status of *EGFR* genes, we compared the expression patterns with and without chromothripsis-like events in *EGFR* mutation-positive cases. As a result, the pathways associated with inflammatory and immune responses were upregulated in chromothripsis-negative cases (**Supplementary Fig. S16**). Chromosomal missegregation and formation of micronuclei during the occurrence of chromothripsis, such as

cGAS-STING pathway, would be censored by an immune surveillance system (Mackenzie KJ et al. 2017 *Nature*), possibly triggering immune response activation. To allow cancer cells to escape from the attacks of the recruited immune cells, the cancer cells may have to change their gene expression programs. Possibly, in parallel with the chromothripsis-like event development, the immune escape-related genes should have been inevitably activated, although further minute experimental validation is needed to validate this hypothesis. We have included this discussion in the revised manuscript (Lines 506–513) and the legend of **Supplementary Figure S16**.

Minor comments:

1. Line 59-60, please spell “SNV” and “SVs” at the first appearance.

We have defined “SNVs” and “SVs” in the revised manuscript (Lines 59–61).

SNVs: single nucleotide variants

SVs: structural variants

2. Line 137, what was the average length of short-read sequencing reads?

We used short read whole-genome sequencing data with 151-bp paired-end. We have added this information in the **Methods** section of the revised manuscript (Line 639).

3. The figures should be modified by using uniform font size and color etc. Please carefully check the grammars throughout the manuscript.

We have modified the figures using uniform font sizes and colors. We have also carefully checked the grammar in the manuscript through the English proofreading.

Point-by-point responses to Reviewer #2:

The authors performed genome-wide haplotype-specific SNV, SV, methylation, and transcriptome analysis in 21 cancer samples and their normal counterpart using Nanopore sequencing and Illumina sequencing. The results were obtained using a number of existing algorithms and some in-house script. Studying cancer genome, epigenome, and transcriptome in a haplotype-specific manner is an important topic and technically feasible with the latest sequencing technologies and bioinformatic algorithms. However, I have following concerns regarding the manuscript writing, methods and results:

1. Most statements in Introduction are lack of support from either literature or experiments of this manuscript. For example, there is no literature citation in statements “A number of important drug targets and biomarkers have been identified, such as EGFR and BRAF.” in page 4. The Introduction section should be completely revised by adding citations to every scientific conclusion.

We are sorry for the lack of citations. We have added appropriate references in the Introduction section. The newly added citations are listed as follows.

- Campbell, P. J. *et al.* Pan-cancer analysis of whole genomes. *Nature* **578**, 82–93 (2020).
- Bailey, M. H. *et al.* Comprehensive Characterization of Cancer Driver Genes and Mutations. *Cell* **173**, 371-385.e18 (2018).
- Ramalingam, S. S. *et al.* Overall Survival with Osimertinib in Untreated, EGFR -Mutated Advanced NSCLC . *N. Engl. J. Med.* **382**, 41–50 (2020).
- Chapman, P. B. *et al.* Improved Survival with Vemurafenib in Melanoma with BRAF V600E Mutation. *N. Engl. J. Med.* **364**, 2507–2516 (2011).
- Pane, F. *et al.* BCR/ABL genes and leukemic phenotype: From molecular mechanisms to clinical correlations. *Oncogene* **21**, 8652–8667 (2002).
- Soda, M. *et al.* Identification of the transforming EML4-ALK fusion gene in non-small-cell lung cancer. *Nature* **448**, 561–566 (2007).
- Kohno, T. *et al.* KIF5B-RET fusions in lung adenocarcinoma. *Nat. Med.* **18**, 375–377 (2012).
- Cameron, D. L., Di Stefano, L. & Papenfuss, A. T. Comprehensive evaluation and characterisation of short read general-purpose structural variant calling software. *Nat. Commun.* **10**, 1–11 (2019).

- Rang, F. J., Kloosterman, W. P. & de Ridder, J. From squiggle to basepair: Computational approaches for improving nanopore sequencing read accuracy. *Genome Biol.* **19**, 1–11 (2018).
- Wenger, A. M. *et al.* Accurate circular consensus long-read sequencing improves variant detection and assembly of a human genome. *Nat. Biotechnol.* **37**, 1155–1162 (2019).
- Sakamoto, Y., Zaha, S., Suzuki, Y., Seki, M. & Suzuki, A. Application of long-read sequencing to the detection of structural variants in human cancer genomes. *Comput. Struct. Biotechnol. J.* **19**, 4207–4216 (2021).
- Tewhey, R., Bansal, V., Torkamani, A., Topol, E. J. & Schork, N. J. The importance of phase information for human genomics. *Nat. Rev. Genet.* **12**, 215–223 (2011).

2-1. Software parameters are not included in the manuscript, which are critical to the conclusions of the study. This includes but not limited to: 1) how to process reads mapped to multiple locations during reads alignment; 2) WhatsHap assumes that the number of haplotypes are fixed. What is the number of haplotypes used in this study? These might affect the observations and conclusions.

We are sorry for the lack of clear explanations. We have included more detailed descriptions for the computational procedures, including the software parameters, in the **Methods** section (**Lines 648–649** and **652**) and **Supplementary Methods**. The essential points are as follows.

1) For the reads mapped to multiple locations during the alignment, low-quality reads with <20 mapping quality, including reads with multiple hits (mapping quality: 0), were filtered out in WhatsHap. We have added this description in the “Haplotype phasing” section of **Supplementary Methods**.

2) First, in this study, we assumed two as the number of haplotypes for the WhatsHap call. We have clarified this in the revised manuscript (**Lines 115–117**) and **Supplementary Methods**. Then, we constructed maternal/paternal haplotypes solely using the “normal” germline SNPs for each case. For the cancer specimens, we just mapped the detected somatic mutations (SNVs and SVs) to the thereby constructed phase blocks. We have added this description in the revised manuscript (**Lines 248–251**) and **Supplementary Methods**. It is also possible that the WhatsHap analysis should be

performed for the cancer specimens considering somatic mutations and aneuploidy. However, at this stage, we consider that the technical noise that should be considered for the analysis would cause more harm than good in cancer specimens where the number of haplotypes generally deviates from two.

2-2. For example, the authors observed that some high depth regions have shorter phased blocks. If a region has multiple copies in the genome, some reads from other non-identical copies might be mapped to this region, this will make the data look like having more than two haplotypes. If the number of haplotypes in WhatsHap is set to 2, the algorithm might be confused and reports shorter phased blocks.

We assume this issue is related to the analysis shown in **Figure 2**. There, we compared the “average” sequencing depth with block length in each specimen. In this analysis, we found some specimens had short phased blocks, even at a “high” average sequencing depth. Originally, we had considered that the sequencing depth should not be a major factor to determine the block length. However, according to the reviewer, high CN aberrations could affect the length of phased blocks. To address this issue, we compared the CN profiles from Illumina short read sequencing data with the length of the phased blocks. As a result, the CN profile seemed not to affect the length of phased blocks. We have included this result in the revised manuscript (**Lines 185–187, Supplementary Fig. S2**).

3. The authors claim that they obtained about 0.2Mb ~ 8.5 Mb long phased SNV blocks using ~20kb long nanopore reads. These numbers are reasonable for phasing germline SNVs. However, I am not convinced that somatic SNVs with distance longer than reads can be unambiguously phased according to the described methods. The key difference between phasing germline SNVs and phasing somatic SNVs is that the number of haplotypes is known for phasing germline SNVs but unknown for phasing somatic SNVs.

When the number of haplotypes is known, for example in the case of phasing germline

SNVs in a diploid genome, assuming that the reads mapping is unique and perfectly correct, a phased block can be unambiguously extended as long as there are a couple reads spanning two heterozygous SNVs. Thus, it is feasible to obtain phased blocks much longer than reads (Figure C1).

When the number of haplotypes is unknown, for example in the case of phasing somatic SNVs in cancer, microbiome, or highly mutable virus, a phased block cannot be unambiguously extended (Figure C2).

First, we thank the reviewer for precisely understanding the procedure despite our poor explanations. As we described in response 2-1, to avoid any confusion, we have clearly stated that we did not construct the phased blocks using cancer mutations. We constructed them only by considering germline SNPs and just mapped the somatic mutations to the constructed phased blocks. In theory, the approach suggested by the reviewer is possible. However, we refrained from taking that approach because it may induce some technical errors. In addition, some important issues, such as the occurrence order of the somatic mutations, would not be delineated once the cancer genomes were phased regardless of the varying values of VAFs at individual mutation sites. We have included this discussion to the revised manuscript (Lines 546–551).

The methods used in this manuscript phase germline SNVs first and then assign somatic SNVs to the inferred haplotype by looking at the linkage between somatic SNVs and their neighboring germline SNVs. This can NOT phase multiple somatic SNVs with distance longer than reads as explained in Figure C1 and Figure C2 since the samples might have more than two haplotypes induced by somatic SNVs. The authors should either phase distant somatic SNVs using better algorithms/technologies or acknowledge that somatic SNVs with distance longer than reads actually can NOT be phased unambiguously. The current writing in Results should be also revised because it gives readers the incorrect impression that somatic SNVs with distance longer than reads can be phased.

Phasing SVs has the same problem.

Then, we thought this was an essential issue. By the careful explanation of the reviewer, we understand that the phasing of cancer mutations cannot go beyond the read lengths of the individual reads, especially where the aneuploidy should also be considered. To avoid any confusion, we have carefully modified the descriptions (**Lines 248–251**). We have also included the (redrawn) Figures C1 and C2 in **Supplementary Figure S6**. Nevertheless, we found that, at least in some cases, the somatic mutations could be phased using the individual long reads as supporting evidence (**Fig. 3f**). This analysis was particularly useful when the occurrence order of the cancerous mutations and clonal structure of the cancer cells were analyzed. This analysis should be further integrated with the phylogenetic analysis, such as PyClone (also see Responses 1-1 and 1-2). To demonstrate the case of further extensive analysis in this direction, we have also included these descriptions to the revised manuscript (**Lines 307–317**).

For further future perspectives, to resolve this issue more generally, challenges should be started to associate multiple mutations beyond the read length. Particularly in cancers, mutations should have been continuously accumulating during the cancer evolution in a time-lapse manner. If two mutations occurred at different time points, both single and double mutant cells could be included in the given bulk tissue. Future long read phasing tools have to explicitly handle such a complex feature of cancerous mutations. Obviously, the number of haplotypes should not be assumed to be two, considering the aneuploidy of cancer genomes, which may also be variable depending on different genomic regions. For the experimental validations, single-cell long read DNA sequencing is needed. We have added this discussion to the **Discussion** session (**Lines 551–555**).

4. The authors used Nanopolish to detect DNA methylation. Although a couple of other tools can report haplotype-specific methylation, Nanopolish does NOT do it. The authors should clarify how they obtain haplotype-specific methylation or acknowledge that the methylation profiles they obtained are not haplotype-specific.

We are sorry for this confusion. We used nanopolish to call CpG methylation statuses in each read. Then, using haplotype information on each read from “WhatsHap haplotag,”

we classified the reads with the methylation information into two haplotypes. We calculated methylation frequencies in each haplotype category using the given script in nanopolish. We have added this description in the **Methods** section (**Lines 739–742**) and **Supplementary Methods**.

5. The haplotyping results using RNA-seq reads harboring multiple SNPs were compared to haplotype inference using gDNA sequencing data. This is plausible in general, but the impact of RNA editing, which might affect the results, is not discussed.

Thank you for this careful comment. We counted overlaps between the possible A-to-I RNA editing sites and the A>G SNPs using in haplotyping of RNA-seq (**Supplementary Table S8**). As a result, 0.22%–0.55% of A>G SNPs to total phased SNPs were overlapped. The result indicates that the impact of RNA editing should be small if any. We have added this discussion in the revised manuscript (**Lines 339–343**).

REVIEWERS' COMMENTS

Reviewer #1 (Remarks to the Author):

Appreciate authors's efforts to address my comments.

I dont have further comments.

Reviewer #2 (Remarks to the Author):

All my comments have addressed in the revised manuscript.

Point-by-point responses for the 2nd revision

Point-by-point responses to Reviewer #1:

Appreciate authors's efforts to address my comments.

I dont have further comments.

We thank the reviewer for the comment.

Point-by-point responses to Reviewer #2:

All my comments have addressed in the revised manuscript.

We thank the reviewer for the comment